# Neural Manifold Regularization: Aligning 2D Latent Dynamics with Stereotyped, Natural, and Attempted Movements

## Abstract

Mapping neural activity to behavior is a fundamental goal in both neuroscience and brain-machine interfaces. Traditionally, at least three-dimensional (3D) latent dynamics have been required to represent two-dimensional (2D) movement trajectories. In this work, we introduce Neural Manifold Regularization (NMR), a method that embeds neural dynamics into a 2D latent space and regularizes the manifold based on the distances and densities of continuous movement labels. NMR pulls together positive pairs of neural embeddings (corresponding to closer labels) and pushes apart negative pairs (representing more distant labels). Additionally, NMR applies greater force to infrequent labels to prevent them from collapsing into dominant labels. We evaluated NMR across four modalities of neural signals and three types of movements. When combined with a linear regression decoder, NMR outperformed other dimensionality reduction methods by over 50% across 68 sessions. The highly consistent neural manifolds extracted by NMR enable robust motor decoding across sessions, years, and subjects using a simple linear regression decoder. Our code is uploaded.

## 1 Introduction

Ongoing breakthroughs in neural recording technologies have led to an exponential increase in the number of simultaneously recorded neurons. To interpret this high-dimensional neural data, manifold analysis has emerged as a promising population-level technique in both neuroscience (Cunningham & Yu, 2014; Jazayeri & Ostojic, 2021) and cognitive science (Beiran et al., 2023; Jurewicz et al., 2024). Analyzing neural manifolds helps to illuminate representations in both biological (Gardner et al., 2022; Hermansen et al., 2024) and artificial (Cohen et al., 2020; Chung & Abbott, 2021; Wang & Ponce, 2021; Dubreuil et al., 2022) neural networks. Because neural population dynamics are high-dimensional, dimensionality reduction methods are necessary to visualize low-dimensional latent dynamics. However, there is a trade-off between representation capacity and dimensionality.

Classical dimensionality reduction methods like principal components analysis (PCA) require eight to fifteen dimensions to represent a simple and stereotyped eight-direction center-out reaching task (Gallego et al., 2020; Gallego-Carracedo et al., 2022). Using the same dataset, state-of-the-art (SOTA) dimensionality reduction methods achieve even better performance using only four dimensions (Zhou & Wei, 2020; Schneider et al., 2023). However, since only 3D spaces are directly visible, these studies have to either display the four dimensions in two separate figures (Zhou & Wei, 2020) or manually remove one dimension (Schneider et al., 2023) to visualize the data. In both cases, further reducing the dimensionality of these low-dimensional latent dynamics is necessary. In a 3D latent space, eight groups of latent dynamics are clearly visible. Unfortunately, the reaching trajectories cannot be identified from the latent dynamics, even when the latent dynamics are trained to align with reaching trajectories (Schneider et al., 2023).

Many hand movement trajectories, such as center-out reaching, random target reaching (O'Doherty et al., 2017; Lawlor et al., 2018), and handwriting (Willett et al., 2021), occur within a 2D physical space. Arguably, the ultimate goal of dimensionality reduction methods is to reveal—either unsupervised or supervised—2D latent dynamics that are well-aligned with, or even indistinguishable from, movement trajectories. However, a 2D latent space has significantly less representational ca-

pacity than a 3D latent space. For body movements within 2D physical spaces like open field arenas, W-shaped mazes, figure-8 mazes, or radial arm mazes, previous dimensionality reduction methods such as Uniform Manifold Approximation and Projection (UMAP) (McInnes et al., 2018) require a 3D latent space to avoid overlap in their latent dynamics (Gardner et al., 2022; Tang et al., 2023; Yang et al., 2024). To our knowledge, no studies have demonstrated the successful use of 2D latent dynamics to represent 2D movement trajectories.

Here, we focus on neural-behavioral analysis, particularly hand movements, which have been extensively studied. We chose hand movement tasks as a testbed for dimensionality reduction methods because: 1) multi-channel recordings provide the necessary high-dimensional data for dimensionality reduction, 2) the diversity of hand movement tasks enables testing different types of task labels, 3) long-term recordings across months and years allow for testing model consistency, 4) a variety of neurophysiological signal types are available, and 5) public open-source datasets enable benchmarking across models. We extended our method to body movement tasks to assess its generalizability.

## 2 RELATED WORK AND OUR CONTRIBUTIONS

There are at least **five categories** of dimensionality reduction methods:

Linear methods: These include techniques like PCA, jPCA (Churchland et al., 2012), demixed PCA (dPCA) (Kobak et al., 2016), and preferential subspace identification (PSID) (Sani et al., 2021). PCA captures the majority of variance in the data, jPCA reveals rotational dynamics in monkey reaching, dPCA further isolates task-related components, and PSID can extract latent dynamics that predict motion during reach versus return epochs.

Nonlinear methods: Techniques such as UMAP and t-distributed stochastic neighbor embedding (t-SNE) (Van der Maaten & Hinton, 2008) are widely used in biological data, such as identifying different neuron cell types (Lee et al., 2021). While these methods can reveal distinct identities, they often collapse temporal dynamics that resemble neural activity. UMAP, when combined with labels, has been used for dimensionality reduction (Schneider et al., 2023; Zhou & Wei, 2020).

Generative methods using recurrent neural networks (RNNs): Models such as fLDS (Gao et al., 2016), latent factor analysis via dynamical systems (LFADS) (Pandarinath et al., 2018), AutoLFADS (Keshtkaran et al., 2022), and RADICaL (Zhu et al., 2022) have been shown to better model single-trial variability in neural spiking activity compared to PCA. However, these methods often rely on restrictive explicit assumptions about the underlying data statistics.

Label-guided generative methods using variational autoencoders (VAEs): Methods such as Poisson identifiable VAE (pi-VAE) (Zhou & Wei, 2020), SwapVAE (Liu et al., 2021), and targeted neural dynamical modeling (TNDM) (Hurwitz et al., 2021; Kudryashova et al., 2023) fall into this category. For instance, pi-VAE uses eight reaching directions as labels to structure the latent embeddings, resulting in eight well-separated latent dynamics in M1.

Contrastive learning methods: Recently, contrastive learning has been introduced for learning robust, generalizable representations of neural population dynamics. Examples include CEBRA (Schneider et al., 2023) and Mine Your Own vieW (MYOW) (Azabou et al., 2021). When trained with hand trajectories, CEBRA demonstrates the most disentangled latent dynamics compared to pi-VAE and AutoLFADS; however, these latent dynamics are not aligned with the actual hand trajectories.

**Our specific contributions are as follows:**

1. Introduction of NMR: A dimensionality reduction method that regularizes latent neural embeddings based on label distances and densities. NMR leverages the continuous nature of movement labels to extract disentangled neural manifolds and addresses label imbalance by applying a pushing force inversely related to the frequency of rare labels. NMR is the first to address imbalanced labels for time-series neural data, and also the first to do so without adding or removing any data samples.

2. Simplification of contrastive regularizer (ConR) loss: NMR replaces the NCE (noise-contrastive estimation) loss used in the CEBRA (Schneider et al., 2023) with a significantly simplified version of the ConR loss (Keramati et al., 2023). The original ConR loss involved six hyperparameters that required fine-tuning for each session. Our modified ConR loss simplifies this by reducing it to a

single temperature hyperparameter. While the original ConR loss showed marginal improvements of less than 5% over previous models, our modified version outperforms CEBRA by over 50%.

3. Evaluation across modalities and movements: We evaluate NMR against CEBRA and pi-VAE using four modalities of neural signals and three types of movements. No previous studies have evaluated dimensionality reduction techniques on LFP signals or attempted to visualize latent dynamics during attempted movements. NMR outperforms the other methods under all conditions.

4. Stability and generalizability across time and monkeys: We assess the stability of our models across months using the same training parameters, as well as their generalizability across monkeys. NMR demonstrates the highest stability over time and superior motor decoding performance across monkeys, even when using the same set of parameters.

## 3 MODEL

### 3.1 MOTIVATION: CONTINUOUS AND IMBALANCED LABELS IN CONTRASTIVE LEARNING

Contrastive learning involves three types of samples: an anchor (or reference sample), positive samples, and negative samples. Positive samples, also known as augmented samples, share the same label as the anchor but are generated by applying transformations to the anchor, such as rotation, flipping, cropping. For time-series data, such as neural dynamics, positive (or augmented) samples are often created by selecting time-offset samples from the anchor, preserving temporal relationships. The goal of contrastive learning is to train the model to bring positive samples closer to the anchor in the latent space while pushing negative samples farther away, effectively learning representations that capture meaningful similarities and distinctions.

The contrastive learning-based method CEBRA outperforms other dimensionality reduction techniques for neural-behavior data analysis. However, it has two key limitations when applied to continuous behavioral data, such as movements. First, CEBRA does not take advantage of the fact that movements are continuous; instead, it treats movement locations or velocities as discrete classes, similar to how images are handled (Fig 1b, left). Second, CEBRA fails to account for the highly imbalanced distribution of movement positions or velocities (Fig 1a-c). In each reach trial, velocities are near zero, and hand positions are close to the center (0, 0) at the start and end of movements, while large velocities or distant hand positions are rare. Such imbalanced distributions are common in real-world data (Yang et al., 2021) and differ significantly from manually curated and balanced datasets like ImageNet (Deng et al., 2009). In the neuroscience field, **previous studies have either neglected the issue of imbalanced labels or downsampled the frequent labels** (Appendix A.1).

### 3.2 MODIFIED AND SIMPLIFED LOSS FUNCTION FROM CONR

Our loss function was modified from the original ConR, which has six hyperparameters. First, there is the temperature $\tau$, used for regularizing feature similarity, which we retained. Second, the distance threshold $\omega$ determines whether paired samples are positive or negative; we replaced this with the median value of pairwise distances (Fig 1e). Third, the pushing power $\eta$, which was manually assigned in their code for all datasets, was replaced with the inverse frequency of the sample distribution in our implementation. Fourth, the hyperparameter $e$ was used for regularizing label distance. It was mentioned only in the code and not in the paper. We retained $e$ and found that it could be assigned the same value as the temperature $\tau$. Fifth and sixth were $\alpha$ and $\beta$, which were used for regularizing the regression and contrastive losses, respectively. Since we did not compute the regression loss, we removed these two hyperparameters. In summary, we only used $\tau$ and $e$, and our model performed well and robustly across the 68 sessions of data we evaluated.

Our NMR model utilizes the same feature encoder as CEBRA, ensuring that the extracted neural embeddings are identical in both models (Appendix A.2). To integrate the ConR loss into CEBRA, we also modified the data sampling strategy. In CEBRA, each training epoch consists of three batches of samples: anchor, positive, and negative. The positive batch is created with a fixed time offset (e.g., 1 or 10 ms) from the anchor, while the negative batch is uniformly sampled from the entire time series. To compute the ConR loss, we utilize the same anchor and positive batches extracted by CEBRA. **The samples in the positive batch will be classified as positive, negative, or discarded (Fig 1d), depending on the difference between the ground truth and predicted**

labels, as well as the threshold for label distance (details provided in the next section). While CEBRA only requires continuous labels once to determine the indices of the positive batch, NMR retains the continuous labels and reuses them in the ConR loss. The negative batch is no longer needed. It is important to note that NMR does not alter the neural embeddings or labels, nor does the modified sampling strategy introduce any additional neural data or labels.

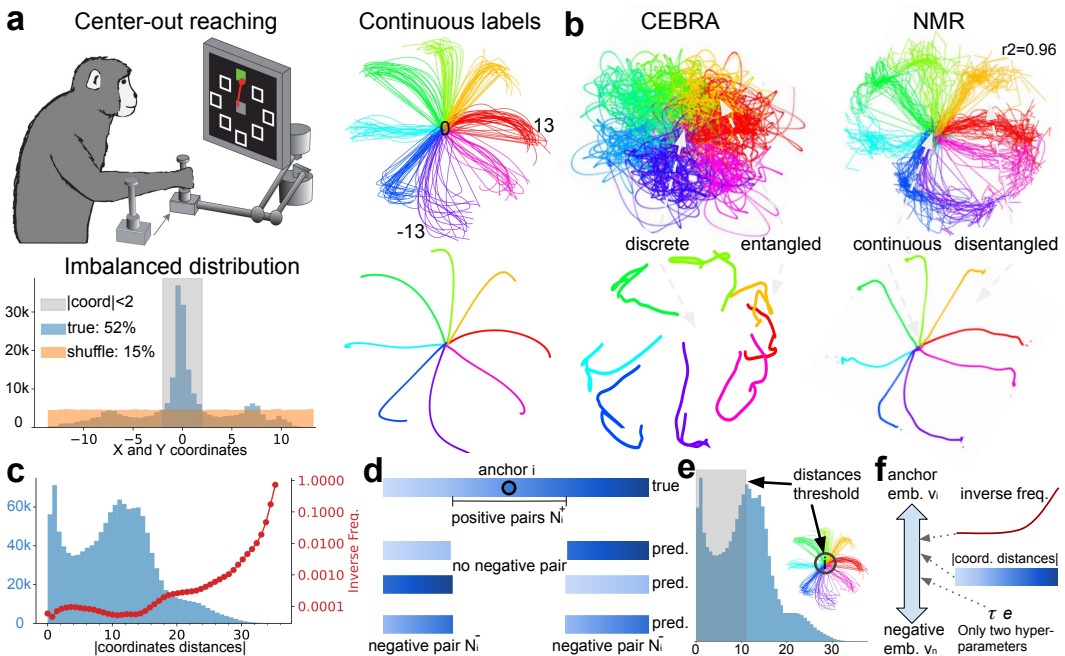

Figure 1: NMR introduces a novel loss function to map 2D latent dynamics with 2D stereotyped hand movements. **a** A monkey performs a center-out reaching task in eight equally spaced directions (modified from Perich et al. (2018)). The slower speed at the beginning of the movement and the central starting point contribute to a highly imbalanced distribution of coordinates around (0, 0). **b** CEBRA extracts latent dynamics that are misaligned with movements (original figures). In contrast, NMR extracts latent dynamics that are closely aligned with movement trajectories, making them nearly indistinguishable. **c** The count (Y-axis, left) and inverse frequency (Y-axis, right) of pairwise distances between X and Y coordinates. Only 10 percent of the coordinates from the figure above are shown. **d** Smooth gradients of blue represent continuous labels. **e** The distance threshold is set to the median of all absolute coordinate distances. **f** The pushing force in the feature space is determined by the inverse frequency of label distances, the label distances, and two hyperparameters.

## 3.3 New Loss function for CEBRA

NMR predicts the labels of anchor samples through linear regression, using their embeddings and corresponding ground truth labels, without altering the embeddings or introducing new labels. Fig 1d illustrates how positive and negative pairs are selected based on true labels (1st row), predicted labels (2nd to 4th rows), and the distance threshold (horizontal line below the 1st row) (Appendix A.3). Samples with distances to an anchor below a specified threshold (1st row, colorbar within the horizontal line) are classified as positive pairs, regardless of their predicted labels. Samples far from the anchor (2nd to 4th rows, six colorbars outside the horizontal line) are either discarded (2nd and 3rd rows) or classified as negative pairs (4th row), depending on their predicted labels. Samples in the 2nd and 3rd rows are discarded because their predicted labels (represented by very dim or dark blue colors) are far from the anchor, irrespective of whether the prediction is correct (2nd row) or incorrect (3rd row). In contrast, samples in the 4th row are considered negative pairs because their predicted labels (medium blue) are closer to the anchor than the threshold, i.e., distant samples have been mispredicted as nearby samples. Similar to the original ConR loss, the label distance is calculated using the $L1$ distance, which is the sum of the absolute differences between the X-coordinates, Y-coordinates, and hand reach angles of any paired labels. **Although our initial**

**sampling approach mirrors that of CEBRA, during the computation of the ConR loss we improve negative sampling by selecting samples supervised by labels, and we improve positive sampling by filtering out unintended negative samples** (Fig 8, Appendix A.4).

Let $d(\cdot, \cdot)$ represent the distance measure between two labels. The ground truth sample label is $y$ and predicted sample label is $\hat{y}$. For each anchor sample $i$, the positive samples are those that satisfy $d(y_i, y_p) < \hat{d}$, the negative samples are those that satisfy $d(y_i, y_n) > \hat{d}$ and $d(\hat{y_i}, \hat{y_n}) < \hat{d}$, where $\hat{d}$ is the median of all pairwise distance shown in Fig 1e.

Let's denote $v_i$, $v_p$, and $v_n$ as the neural embeddings of corresponding true labels of $y_i$, $y_p$, and $y_n$. $N_i^+$ is the number of positive samples, $N_i^-$ is the number of negative samples. $K_i^+ = \{v_p\}_p^{N_i^+}$ is the set of embeddings from positive samples, $K_i^- = \{v_n\}_n^{N_i^-}$ is the set of embeddings from negative samples. $sim(\cdot, \cdot)$ is the similarity measure between two feature embeddings (e.g. negative $L_2$ norm). For each anchor $i$ whose neural embedding is $v_i$, true label is $y_i$, and loss is:

$$\mathcal{L} = \frac{1}{N_i^+} \sum_{v_j \in K_i^+} -\log \frac{\exp(sim(v_i, v_j)/\tau)}{\sum_{v_p \in K_i^+} \exp(sim(v_i, v_p)/\tau) + \sum_{v_n \in K_i^-} S_{i,n} \exp(sim(v_i, v_n)/\tau)} \quad (1)$$

where $\tau$ is a temperature hyperparameter and $S_{i,n}$ is a pushing weight for each negative pair shown in Fig 1f:

$$S_{i,n} = \frac{1}{p_{d(y_i, y_n)}} exp(d(y_i, y_n)e) \quad (2)$$

where $\frac{1}{p_{d(y_i, y_n)}}$ is the inverse frequency of labels distances distribution shown in Fig 1c. Both exponential and linear label distances achieve similar results, with $e$ serving as a hyperparameter to scale the label distance. The final loss is the summed loss $\mathcal{L}$ over all anchors $i$ (Appendix A.7).

### 3.4 Ablation studies and comparison with supervised methods

The performance gain of NMR over CEBRA (Fig 1b) can be attributed to two factors. First, NMR uses multiple positive pairs ($K_i^+$), whereas CEBRA uses only a single positive pair. Second, the pushing weight $S_{i,n}$ for the negative pairs is scaled based on their distances. We conducted ablation studies (Fig 9a-d) and found that using multiple versus one positive pair has negligible improvement on the alignment of latent dynamics and decoding performance. In contrast, when the pushing weight is set to one, the latent dynamics are squeezed—that is, large but infrequent values are collapsed into small but frequent values. This is precisely what NMR aims to resolve. Therefore, it is the pushing weight $S_{i,n}$ applied to the negative pairs that contributes to the improved performance.

NMR explicitly trains the latent dynamics to align with movements (Appendix A.8). An alternative end-to-end approach involves training a deep neural network to directly predict movements from neural data, with its latent dynamics implicitly regularized during this process. To explore this, we trained a long short-term memory (LSTM) network, which is specifically designed for modeling time-series neural data. However, the LSTM's performance was inferior to NMR's (Fig 10).

We benchmarked the motor decoding performance of NMR against SOTA methods utilizing transformer and other architectures, including NDT1 (Ye & Pandarinath, 2021), EIT (Liu et al., 2022), NDT2 (Ye et al., 2023), and POYO (Azabou et al., 2023). We report results from prior studies where models were trained from scratch using 80% or 90% of data from a single session and tested on the remaining 20% or 10% holdout data from the same session (Appendix A.10). NMR outperforms all previous models in the same session and across 35/37 sessions over ten months.

## 4 Experiments

Two common ways to evaluate dimensionality reduction methods are: (1) the qualitative direct visualization of the revealed latent dynamics, and (2) the quantitative decoding performance of task variables using a decoder. The decoding performance is measured by the explained variance (r²) between the ground truth and the decoded movement trajectories. Although better decoding performance can be achieved with complex decoders, we choose to enforce a linear mapping across the three methods to prevent excessively complex decoders from compensating for poor latent dynamics

estimation (Pei et al., 2021). Nevertheless, since all three methods are trained using movement labels, motor decoding performance—both within the same session and across different sessions and subjects—is highly relevant for practical applications such as the brain-machine interfaces.

We evaluated NMR against the supervised deep learning-based models CEBRA and pi-VAE. These models were chosen because they (1) represent two categories—contrastive and generative—of dimensionality reduction methods that have achieved SOTA performance; (2) have released their code and use publicly available datasets; and (3) benchmark against previous models such as PCA, UMAP, fLDS, LFADS, AutoLFADS, and others. **We evaluated all three models using the same neural data and movement labels** (see Appendix A.5 and Table 1 for training parameters). To eliminate bias from using data from a single session in a single brain area—where pi-VAE and CEBRA were previously tested—we conducted experiments across a total of 68 sessions (Appendix A.6). These experiments involved neural signals from four modalities: M1, PMd, and S1 in monkeys, and the precentral gyrus in humans. Importantly, we included three different movement tasks. While our primary focus is on hand movements, we also evaluated body movements using neural data from the rat hippocampus. The results demonstrated a 37% improvement of NMR over CEBRA (Fig 11).

### 4.1 NMR EXPLAINS THE LARGEST AND MOST CONSISTENT MOVEMENT VARIANCE

Our initial focus was on classical stereotyped center-out reaching tasks, similar to the task in Fig 1, but with neural data from the motor cortex (M1) and premotor cortex (PMd) instead of the somatosensory cortex (S1). We found that NMR significantly outperformed hyperparameter-optimized CEBRA and pi-VAE models by a large margin (M1: 0.88 vs 0.48 vs 0.43; PMd: 0.9 vs 0.53 vs 0.37, median values, Fig 2). The performance difference between NMR and CEBRA was statistically significant (M1, t = 14.9, p = 6.3e-10; PMd, t = 16.8, p = 1e-8; paired t-test with multiple comparisons correction), as was the difference between NMR and pi-VAE (M1, t = 9.7, p = 2.4e-7; PMd, t = 9.8, p = 2.8e-6). Importantly, NMR exhibited less variability across sessions (M1, 0.03; PMd, 0.02, standard deviation) compared to both CEBRA (M1, 0.1; PMd, 0.06) and pi-VAE (M1, 0.18; PMd, 0.18). Multiple runs with different parameters within the same session showed that CEBRA is more

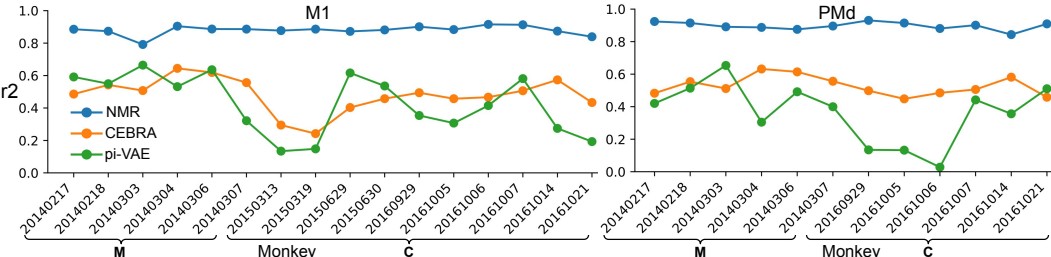

Figure 2: NMR consistently outperforms CEBRA and pi-VAE across different brain areas, monkeys, and hemispheres. The Y-axis displays the explained variance, while the X-axis shows the session dates (formatted as YYYYMMDD) for 16 sessions in M1 and 10 sessions in PMd. See Table 2 for details about each session. Task labels represent hand velocity. The best hyperparameters were chosen when evaluating the CEBRA and pi-VAE models. Model parameters were kept fixed across all 28 sessions. Figs 1213 illustrate the hyperparameter search and stability of the CEBRA and pi-VAE models, respectively, while Fig 14 shows the results using 3D CEBRA and pi-VAE models.

robust than pi-VAE (Figs 1213), consistent with previous findings from the CEBRA paper. Since CEBRA and pi-VAE typically perform better at higher dimensionality, we also compared 2D NMR with 3D CEBRA/pi-VAE (i.e., without further dimensionality reduction using PCA on the original 3D output). The results remained similar (Fig 14). In summary, NMR explained the largest variance of hand movements and demonstrated the most consistent performance across sessions.

### 4.2 DECODING WITHIN AND ACROSS SESSIONS, SUBJECTS, AND YEARS

Since NMR explains the largest movement variance ($r^2$) across all sessions in both M1 and PMd, we further investigated whether the latent dynamics aligned with movements in one session could be utilized to decode movements in other sessions or even across different subjects (Appendix A.8).

Fig 3 shows the within-session decoding performance (values on the diagonal) and cross-session decoding performance (values off the diagonal) for the three models. Consistent with the explained variance results, NMR significantly outperformed CEBRA (t = 11.5, p = 2.4e-8, paired t-test with multiple comparisons correction) and pi-VAE (t = 6.2, p = 5e-5) in decoded variance within sessions. The performance gap was even more pronounced for cross-session decoding, with NMR performing nearly twice as well as CEBRA (t = 18.5, p = 1.5e-47) and six times better than pi-VAE (t = 21, p = 1.4e-55). Additionally, CEBRA almost tripled the performance of pi-VAE (t = 9.6, p = 3.6e-18). These results are consistent with the smaller cross-session standard deviation observed in the Fig 2.

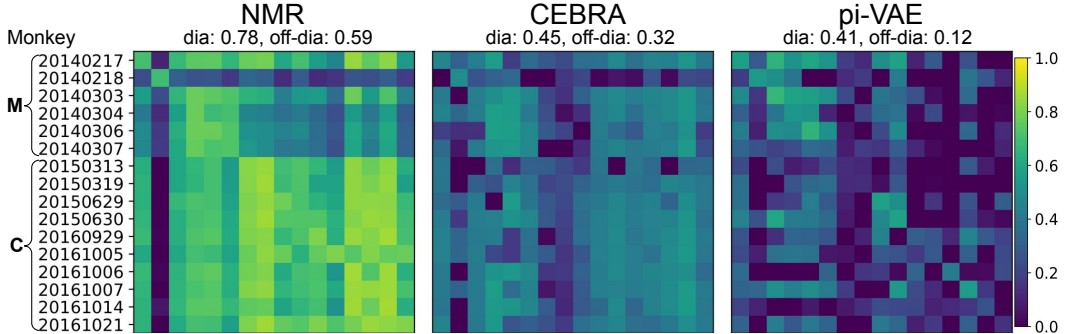

Figure 3: Within- and across-session movements decoding performance (r²) in M1 for Monkey M and C. Fig 15 shows the decoding results in PMd. Appendix A.9 shows the technical details.

Interestingly, we did not find a causal relationship between the variability of decoding performance and the number of neurons or trials in each session (Table 2). The variability is unlikely due to neural signals, as the within-session decoding performance of Monkey M on 20140218 is similar to that of five other sessions. The variability is highly likely due to movement changes, as Monkey M on 20140218 had the worst cross-session decoding performance in both M1 and PMd (Fig 15), despite having more neurons than on 20140307 (M1: 38 vs. 26; PMd: 121 vs. 66). In summary, the low-dimensional, high-performance, and stable movement-aligned latent dynamics revealed by NMR enable effective neural decoding across sessions and even across different subjects.

## 4.3 DIMENSIONALITY REDUCTION USING BANDS OF LOCAL FIELD POTENTIAL SIGNALS

Dimensionality reduction methods have predominantly been evaluated on single-neuron data, either through neurophysiological recordings or calcium imaging. However, numerous studies have demonstrated that local field potential (LFP) signals contain movement-related information and can achieve comparable decoding performance to single-neuron data. To explore this further, we tested three models using the LFP signals that accompanied the previous single-neuron recordings.

We first examined whether different bands of LFP signals were modulated by movement (Fig 4a). As expected, movement onset, occurring approximately 300 ms after the go cue, evoked amplitude changes in several LFP bands. Notably, LFP bands across different channels showed distinct modulations, a prerequisite for population decoding and for revealing latent dynamics from high-dimensional neural data. The local motor potential (LMP), which consists of unfiltered and smoothed LFP signals, exhibited the most diverse movement modulation across all channels. We then evaluated the explained variance (Fig 4b) and decoding performance (Fig 17) of NMR and CEBRA across 28 sessions in three representative LFP bands.

The results showed that performance was LFP band-dependent: the LMP and high-frequency band (200-400 Hz) significantly outperformed the middle-frequency band (12-25 Hz). Furthermore, NMR outperformed CEBRA across all three bands—LMP (0.79 vs 0.46), Gamma (0.74 vs 0.44), and Beta (0.36 vs 0.22)—with statistically significant differences (t = 6.8, 7.8, and 3.1; p = 1.8e-5, 3.8e-6, and 0.002, paired t-test with multiple comparisons correction) in both M1 and PMd. However, we observed some variability. NMR's performance dropped below CEBRA in certain bands and sessions (e.g., LMP in Monkey C, 20161006, M1). In contrast to the results with single-neuron data, NMR showed greater variability across sessions (0.15 vs 0.11, 0.2 vs 0.09, 0.23 vs 0.17). Despite this, the overall performance of LFP signals was only slightly lower than that of single-neuron

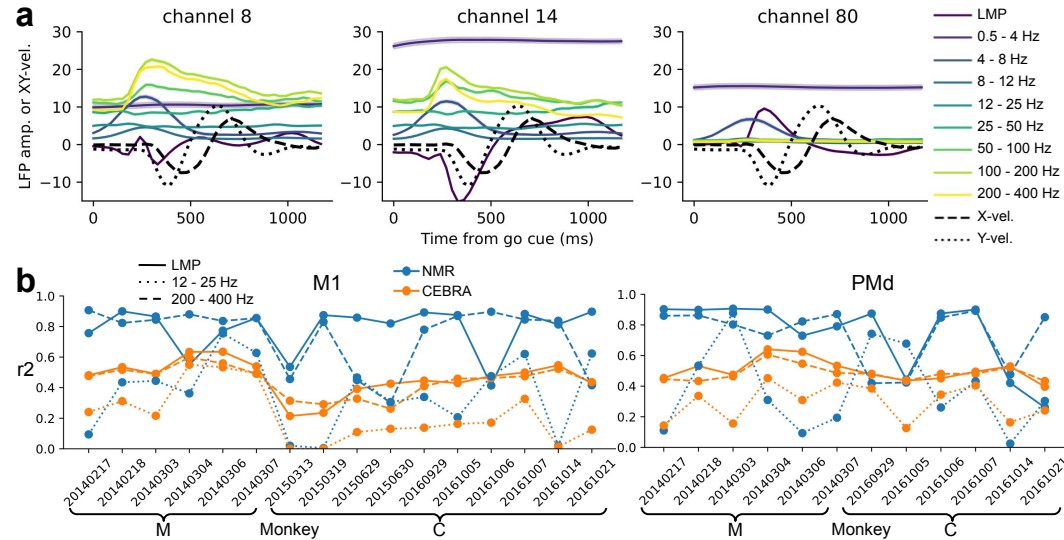

Figure 4: Dimensionality reduction on LFPs. **a** Seven LFP bands along with X- and Y-velocity in three example channels. Error bars represent the standard error of the mean across all trials in this session (Monkey C, 20161014, M1). **b** The explained variance (r²) of the model is shown across all sessions in M1 (left) and PMd (right) for three LFP bands: LMP, 12-25 Hz Beta band, and 200-400 Hz Gamma band. Figs 16 and 17 show the hyperparameter tuning of the two models and the decoding performance on test trials, respectively.

data. In summary, NMR outperforms CEBRA even when using LFP signals, though it exhibits more variability across sessions.

## 4.4 DIMENSIONALITY REDUCTION USING SINGLE-NEURON AND UNSORTED EVENTS

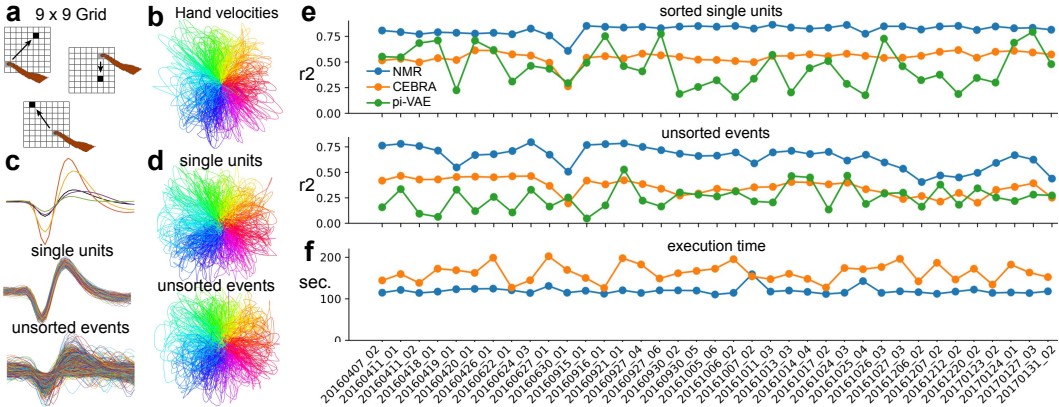

Figure 5: Dimensionality reduction on natural movements using data from single units and unsorted events. **a** Three example movement trials in a 9 x 9 grid on a computer screen (modified from Keshtkaran et al. (2022)). **b** Hand velocities for all reaching movements, with different colors representing different angles. Data are from session indy 20170124 01. **c** Four sorted single units and the remaining unsorted events from one channel. **d** 2D latent dynamics revealed by NMR using both sorted and unsorted data modalities. **e** Explained variance for three models across 37 sessions using sorted single units (top) and unsorted events (bottom). **f** Execution time for NMR and CEBRA, with pi-VAE excluded for comparison since it runs on the CPU instead of the GPU. Figs 181920 show findings with different hyperparameters, decoding performance for test trials with 3D models, and execution time under varying conditions, respectively.

Our previous evaluation, while exhaustive, focused primarily on stereotyped movements. It is important to assess how NMR performs in natural movements without predefined target locations. To address this, we benchmarked three models in a task involving restricted natural movements, where target locations appeared randomly on a 9 x 9 grid on the screen (Fig 5a). In this task, there is no delay period, and trials have variable lengths with almost no overlap in movement trajectories (Fig 5b). Each recording channel contained one or more sorted single units as well as unsorted remaining events (Fig 5c). Surprisingly, both sorted single units and unsorted events were able to uncover movement (velocity)-aligned 2D latent dynamics (Fig 5d).

We benchmarked the three models across 37 sessions over a span of 10 months in one monkey. Consistent with the results from 28 sessions in the center-out reaching task, NMR outperformed CEBRA and pi-VAE by a large margin in all sessions for both sorted single units (0.82, 0.55, and 0.45) and unsorted events (0.65, 0.36, and 0.25) (Fig 5e). Hyperparameter tuning across all 37 sessions for all three models further supported these conclusions (Fig 18). We observed consistent results on the test trials and when using 3D versions of CEBRA and pi-VAE models (Fig 19). Since CEBRA computes the distance between an anchor and all samples in the batch, while NMR does not compute distances for predicted labels that deviate from the true labels, we hypothesized that NMR would have more efficient computing than CEBRA. Supporting this hypothesis, we found that execution time across sessions was significantly shorter for NMR compared to CEBRA, both for single units (119 vs 163 seconds, t = 12, p = 3e-14) (Fig 5f) and for unsorted events (149 vs 166 seconds, t = 3.5, p = 0.001) (Fig 20a). This result held true under different hyperparameters for both models (Fig 20b, c). In summary, NMR demonstrates superior performance for natural movements using data from both single units and unsorted events.

In the previous task, natural movements on a 9 x 9 grid involved unpredictable yet predefined target locations. However, in more realistic scenarios, a target can appear anywhere. To simulate this, we further evaluated the three models on a free natural movements task, where the target could appear at any location on the screen (Fig 6a). NMR revealed 2D latent dynamics that were better aligned with both hand velocity and direction compared to CEBRA (0.88 vs 0.79, Fig 6b). We ran 20 evaluations to compare the performance and stability of the models. Consistent with previous findings, NMR achieved the highest performance (0.79, 0.58, and 0.56) in explaining hand velocities and exhibited the smallest variability across runs (0.002, 0.004, and 0.117) (Fig 6c). Similar trends were observed in the test trials, where NMR showed higher performance (0.77, 0.65, and 0.53) and lower variability (0.005, 0.006, and 0.109) (Fig 21). Additionally, NMR had a shorter execution time compared to CEBRA (146 vs 165 seconds, t = 3.5, p = 0.0025, Fig 21).

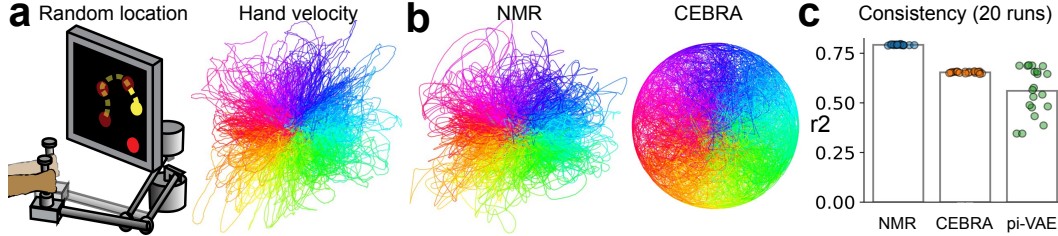

Figure 6: Dimensionality reduction on natural movements with random target locations. **a** A monkey was trained to perform sequences of four reaches to randomly placed target locations (modified from Safaie et al. (2023)). The colors of each reaching trial indicate the angles. **b** 2D latent dynamics revealed by NMR and CEBRA. **c** Explained variance of hand velocities by three models across 20 runs. Fig 21 provides additional details on decoding performance and execution time.

## 4.5 NMR Maps Latent Dynamics to Attempted Center-Out Handwriting

The datasets evaluated so far come from 67 sessions across three different hand-reaching tasks in four macaque monkeys. However, two key questions remain: Can NMR work for attempted or imagined reaching instead of physical hand movements? And how does it perform outside of monkeys? To address these questions, we focused on a dataset involving attempted center-out handwriting in 16 directions by a paralyzed patient. One significant challenge in this task is the absence of measurable hand or finger position data, as the participant must imagine movement trajectories while

following on-screen instructions (Fig 7a). During the task, multiunit threshold crossing data were recorded from the hand knob area. Remarkably, NMR successfully revealed single-trial latent dynamics without any overlap in trials that were 22.5 degrees apart (Fig 7b). The averaged 2D latent dynamics closely matched the imagined movement trajectories ($r^2 = 0.96$, based on hand positions). We optimized the hyperparameters of the three models before evaluating them across 20 runs (Fig 22). Consistent with the results obtained using actual hand positions, NMR also revealed aligned trajectories when trained on hand velocities (Fig 23a). While NMR outperformed both models, CEBRA showed better performance than pi-VAE but still lagged behind NMR (0.78, 0.59, and 0.23, Fig 7c). We observed similar results in the test trials and with the 3D versions of the CEBRA and pi-VAE models (Fig 23b). Consistent with earlier findings, NMR also had a shorter execution time compared to CEBRA (Fig 23c). Overall, NMR reveals the most aligned latent dynamics for attempted handwriting and shows strong potential for applications in brain-machine interfaces.

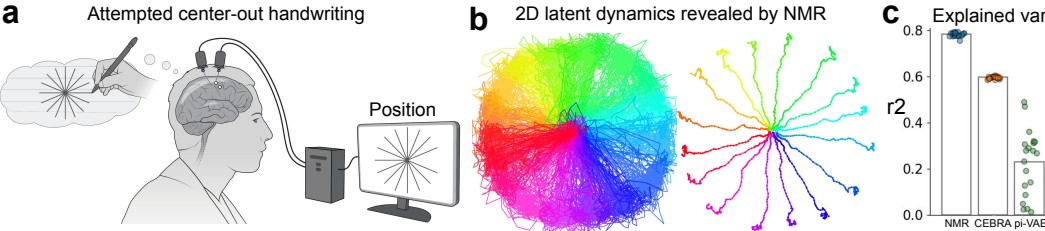

Figure 7: Dimensionality reduction on handwriting attempts in 16 directions. **a** A participant attempted to handwrite in 16 directions, following instructions displayed on a monitor. Neural recordings were made from two 96-channel Utah arrays implanted in the hand knob area of the precentral gyrus (modified from Willett et al. (2021)). **b** Single-trial and trial-averaged latent dynamics were revealed by NMR. **c** Explained variance of hand velocities across three models after 20 runs. Fig 22 shows hyperparameter tuning, and Fig 23 provides further comparison results.

## 5 DISCUSSION

A benchmark of NMR against CEBRA and pi-VAE across multiple brain areas, four modalities of neural signals, and three movement tasks demonstrates NMR's superior performance in uncovering latent dynamics. One of the key strengths of NMR is its ability to extract nearly identical latent dynamics across different brain areas and over extended periods. This capability opens new avenues for both fundamental neuroscience research and brain-machine interface (BMI) applications. Previous studies by Gallego et al. (2020) and Safaie et al. (2023) revealed preserved latent dynamics across time and subjects performing similar behaviors using the PCA method. However, the latent dynamics revealed by NMR (as shown in Figs 1567) are significantly more informative than those uncovered by PCA. We believe NMR will help neuroscientists probe the stability of latent dynamics under various conditions. For BMI applications, we demonstrate that NMR, combined with a simple linear decoder, can predict hand movements across years, subjects, and hemispheres. This capability allows for training latent dynamics within and between subjects, enabling the prediction of movements in other subjects. The linear decoder's lack of hyperparameters is an additional advantage. Furthermore, NMR also revealed almost perfectly aligned 2D latent dynamics in a paralyzed human patient, further highlighting its potential for use in BMI applications for humans.

If the ultimate goal of a dimensionality reduction method is to align latent dynamics with any movements, then NMR is still far from achieving this. For the three movement tasks evaluated in this study, the movement trajectories are relatively simple. For complex movements like handwriting characters such as "m" or "k" (Willett et al., 2021), the latent dynamics will collapse. We believe this is due to the calculation of label distance; geodesic distance might be more suitable than Manhattan or Euclidean distance. Furthermore, we consider speech (Silva et al., 2024)—which involves coordinated movements of the jaw, tongue, lips, and larynx—to be one of the most challenging movement tasks. We believe it is still feasible to reveal the latent dynamics, though they are unlikely to be 2D, if the label distance of articulatory kinematic trajectories (AKTs) (Chartier et al., 2018) can be quantified. A model may need to reduce the dimensionality of both AKTs (coordinated movements in 13 dimensions) and neural dynamics.

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

# A APPENDIX

## A.1 CONTRASTIVE LEARNING IN NEUROSCIENCE

### A.1.1 STUDIES THAT NEGLECT IMBALANCED LABELS

There has been a surge in the development of contrastive learning methods for neural data. Some of these methods do not face class imbalance issues because they address tasks where imbalance is not a concern, such as change point detection (Urzay et al., 2023), generating neural activity to predict behavior (Antoniades et al., 2023), and behavior decoding (Azabou et al., 2021). For cell-type classification, which inherently involves imbalanced classes, previous studies have not explicitly addressed the class imbalance problem. For instance, (Yu et al., 2024) proposed the multimodal NEMO model for cell-type and brain region classification. While they acknowledged the imbalance by reporting macro-averaged F1 scores and balanced accuracy, they did not provide solutions to mitigate the issue. Similarly, (Vishnubhotla et al., 2023) introduced CEED for spike sorting and cell-type classification without mentioning the class imbalance, despite its prevalence in neural cell types. Moreover, other studies have applied contrastive learning to neurophysiological data such as spike sorting without considering class imbalance. (Vishnubhotla et al., 2023) also used CEED for spike sorting without mentioning the class imbalance issue. (Qian et al., 2022) applied contrastive learning for spike sorting but did not consider class imbalance. (Chen et al., 2022) proposed TreeMoCo for neural morphology representation learning but did not consider the inherent imbalance of neural types in the brain.

In summary, most previous studies applying contrastive learning to neuroscience—whether on time-series data or images—do not consider or attempt to address imbalanced labels.

### A.1.2 STUDIES THAT ADDRESS IMBALANCED LABELS

Among the contrastive learning studies in neuroscience that address imbalanced labels, we found all rely on traditional sampling techniques focused on discrete classes. For example: (Dorkenwald et al., 2023) proposed SegCLR for connectomics data. They mentioned: "Examples were rebalanced class-wise by upsampling all classes to match the most numerous classes. During testing, imbalances between classes were balanced by repeating examples from minority classes." (Kostas & Rudzicz, 2020) mentioned: "Wherever there was an imbalance in examples between classes, we under-sampled the majority class(es)..." (Kostas et al., 2021) mentioned: "The P300, ERN, and SSC datasets all had imbalanced class distributions; during training, we adjusted for these imbalances

by undersampling points uniformly of the more frequent classes..." (Shen et al., 2022) mentioned: "The neutral emotion category was not included in the basic version due to an unbalanced number of trials (only four trials for eliciting neutral emotion)."

We are the first to address imbalanced labels for time-series neural data, and also the first to do so without adding or removing any data samples, achieving an 80-100% performance gain over previous SOTA methods.

## A.2 CODE

Operating system: Ubuntu 22.04.3 LTS, GPU: NVIDIA RTX A5000, CPU: Intel Xeon W-2225.

We have uploaded all of our code, including the modified loss function, preprocessing scripts, and figure generation code. We modified only four files in the "cebra" code folder. The latest CEBRA version that we used was released on January 10, so all files except these four have a modified date of January 10. The modified files include two in the data folder (single_session.py and datatypes.py) and two in the solver folder (single_session.py and base.py). We revised three of four files for data sampling (retain continuous labels) as follows:

In single_session.py (Lines 69-71, 76-79), we retained continuous labels for computing the ConR loss later on. Notably, NMR and CEBRA both utilize continuous labels in continuous.py within the "distribution" folder, which we did not modify.

Lines 69-71 for extracting continuous labels and ConR parameters from inputs

```
XY_position = self.continuous_index[:, 0:2]
Z_target = self.continuous_index[:, 2][index.reference]
para = self.continuous_index[0:4, 3]
```

Lines 76-79 for retaining continuous labels

```
index_ref = XY_position[index.reference, 0],
index_pos = XY_position[index.reference, 1],
Z_target = Z_target,
para = para
```

We commented out two lines (Line 75 and 260) because the computation of the ConR loss does not require the embeddings or indices of negative samples extracted by CEBRA.

```
negative=self[index.negative],
negative_idx = reference_idx[num_samples:]
```

Original file for reference:

```
https://github.com/AdaptiveMotorControlLab/CEBRA/blob/main/cebra/
data/single_session.py#
```

In datatypes.py (Lines 55-58 and 64-67 to add labels, and Lines 54 and 63 to comment out negative samples), we used this file to retain the continuous labels, serving a similar purpose as single_session.py. Original file for reference:

```
https://github.com/AdaptiveMotorControlLab/CEBRA/blob/main/cebra/
data/datatypes.py#L46
```

In the solver folder, single_session.py (Lines 72-76 for adding labels and Line 71 for commenting out negative samples) also retains the continuous labels but on the GPU, fulfilling the same function as the files above. Original file for reference:

```
https://github.com/AdaptiveMotorControlLab/CEBRA/blob/main/cebra/
solver/single_session.py#L59
```

The computation of ConR loss is implemented in base.py in the solver folder. This file contains two key parts: Target label prediction (Lines 346-356), where we use linear regression on the CPU with scikit-learn. ConR loss calculation (Lines 61-163), where we use two embeddings, real and predicted labels, and two parameters. Original file for reference:

```
https://github.com/AdaptiveMotorControlLab/CEBRA/blob/main/cebra/
solver/base.py#L225
```

## A.3 INFONCE VS CONR LOSS

The computation of InfoNCE loss contains these key five lines of code:

```
# feature similarity of all positive and negative samples
1. pos_dist = einsum("nd,nd->n", ref, pos)/tau
2. neg_dist = einsum("nd,md->nm", ref, neg)/tau
# attract similar samples
3. pos_loss = -pos_dist.mean()
# repel dissimilar samples
4. neg_loss = logsumexp(neg_dist, dim = 1).mean()
# minimize this loss during in each epoch
5. loss = pos_loss + neg_loss
```

The computation of ConR loss contains these key ten lines of code:

```
# feature similarity of all samples
1. logits = - (features[:, None, :] - features[None, :, :])
        .norm(2, dim=-1).div(t)
# find positive pairs, I_dist is the true pairwise distance,
# w is distance threshold
2. pos_i = l_dist.le(w)
# find negative pairs, p_dist is the the predicted distance
3. neg_i = ((~(l_dist.le(w))) * (p_dist.le(w)))
# feature similarity of positive samples
4. pos = torch.exp(logits * pos_i)
# feature similarity of negative samples
5. neg = torch.exp(logits * neg_i)
# pushing weight
6. pushing_w = inverse_freq * torch.exp(l_dist_XY * e) * neg_i
# denominator (equation on the right)
7. neg_exp_dot = (pushing_w * neg).sum(1)
# denominator
8. loss_single_denom = (pos.sum(1) + neg_exp_dot).unsqueeze(-1)
# single sample ConR loss (numerator/denominator)
9. loss_single = torch.div(pos, loss_single_denom)
# sum and averaged over all samples in the batch
10. loss = (-torch.log(loss_single) * pos_i).sum(1)
        / (pos_i.sum(1)).mean()
```

## A.4 SAMPLING

### A.4.1 NEGATIVE SAMPLING

The key difference between the two losses lies in the selection of negative samples: in NMR, negative sampling depends on behavioral labels, whereas in CEBRA, it is independent of them (Fig 8a). Below, we outline the relevant code and links for negative sampling in CEBRA when supervised with continuous labels.

**Code References for Negative Sampling in CEBRA** The negative sampling process begins in the `ContinuousDataLoader` class:

```
https://github.com/AdaptiveMotorControlLab/CEBRA/blob/main/cebra/
data/single_session.py#L162
```

```
class ContinuousDataLoader(cebra_data.Loader):
    "Contrastive learning conditioned on a continuous labels."
```

The selection of negative indices occurs at the following lines:

```
https://github.com/AdaptiveMotorControlLab/CEBRA/blob/main/cebra/
data/single_session.py#L249
```

```
# Call \sample_prior" function in continuous.py file
# in the \distributions" folder
reference_idx = self.distribution.sample_prior(num_samples * 2)
negative_idx = reference_idx[num_samples:]
reference_idx = reference_idx[:num_samples]
```

The `sample_prior` function is defined in the `Prior` class, which is located in:

```
https://github.com/AdaptiveMotorControlLab/CEBRA/blob/main/cebra/
distributions/continuous.py#L34
```

```
class Prior(abc_.PriorDistribution, abc_.HasGenerator):
    "An empirical prior distribution for continuous datasets."
```

**The `sample_prior` Function**    The `sample_prior` function is responsible for uniformly sampling indices from the dataset. It is defined as follows:

```
https://github.com/AdaptiveMotorControlLab/CEBRA/blob/main/cebra/
distributions/continuous.py#L52
```

```
def sample_prior(self, num_samples: int, offset:
            Optional[Offset] = None) -> torch.Tensor:
    "Return uniformly sampled indices."
    # Generate random integers within the specified range
    return self.randint(self.offset.left, self.num_samples
        - self.offset.right, (num_samples,))
```

The CEBRA paper further explains this process in the Methods/Sampling section, stating: *"In the simplest case, negative sampling returns a random sample from the empirical distribution by returning a randomly chosen index from the dataset."*

**Summary**    It is important to note that most supervised contrastive learning methods guide negative sampling based on labels. However, CEBRA employs an unsupervised negative sampling approach that does not rely on labels. Similarly, NMR does not use labels to guide sampling. Instead, NMR incorporates labels later during the computation of the ConR loss.

A.4.2    POSITIVE SAMPLING

NMR uses the same positive samples extracted by CEBRA. Below, we detail the positive sampling process, referencing the relevant code sections for clarity.

**Sampling Positive Indices**    The indices of positive samples are assigned in the following line within `single_session.py`:

```
positive_idx = self.distribution.sample_conditional(reference_idx)
```

This code is located at:

```
https://github.com/AdaptiveMotorControlLab/CEBRA/blob/main/cebra/
data/single_session.py#L252
```

**The `TimedeltaDistribution` Class**    The `sample_conditional` function is defined in the `TimedeltaDistribution` class within the `continuous.py` file in the `distributions` folder:

```
https://github.com/AdaptiveMotorControlLab/CEBRA/blob/main/cebra/
distributions/continuous.py#L200
```

The `TimedeltaDistribution` class defines a conditional distribution based on continuous behavioral changes over time:

```
class TimedeltaDistribution():
    self.data = continuous  # Continuous movement labels
    self.time_difference[time_delta:] = self.data[time_delta:]
    - self.data[:-time_delta]
    self.index = cebra.distributions.ContinuousIndex(self.data)
```

Here, `self.data` represents the continuous movement labels, and `self.time_difference` calculates the difference over a specified time delta. The `ContinuousIndex` is then initialized with this data.

**The `ContinuousIndex` Class** The `TimedeltaDistribution` class utilizes the `ContinuousIndex` class, defined in `index.py`:

https://github.com/AdaptiveMotorControlLab/CEBRA/blob/main/cebra/distributions/index.py#L131

The `ContinuousIndex` class is responsible for searching the nearest neighbors based on the query data:

```
class ContinuousIndex(distributions.Index):
    def search(self, query):
        distance = self.dist_matrix(query)
        return torch.argmin(distance, dim=0)
```

In this function, `self.dist_matrix(query)` computes the distances between the query points and the dataset, and `torch.argmin` finds the indices of the nearest neighbors.

**The `DistanceMatrix` Class** The `search` function relies on the `DistanceMatrix`, defined in the same `index.py` file:

https://github.com/AdaptiveMotorControlLab/CEBRA/blob/main/cebra/distributions/index.py#L55

The `DistanceMatrix` class implements a naive nearest neighbor search by computing the distances between all pairs of points in the dataset:

```
class DistanceMatrix(cebra.io.HasDevice):
    # Implementation details
```

This approach involves a brute-force computation of distances, which, while simple, ensures accurate neighbor identification for small to medium-sized datasets.

**The `sample_conditional` Function** The `sample_conditional` function within the `TimedeltaDistribution` class orchestrates the positive sampling process:

https://github.com/AdaptiveMotorControlLab/CEBRA/blob/main/cebra/distributions/continuous.py#L240

```
def sample_conditional(self, reference_idx) -> torch.Tensor:
    num_samples = reference_idx.size(0)
    # Return random integers
    diff_idx = self.randint(len(self.time_difference),
                (num_samples,))
    # Time-offset to reference as positive samples
    query = self.data[reference_idx]
                + self.time_difference[diff_idx]
    # Call the search function mentioned earlier
    return self.index.search(query)
```

In this function:

- `num_samples` determines the number of samples to generate. - `diff_idx` selects random indices from the time differences. - `query` computes the new data points by adding the time differences to the reference data. - `return self.index.search(query)` finds the indices of the data points closest to the query points, effectively selecting the positive samples.

**Summary**   By utilizing the continuous movement labels and time differences, the positive sampling process selects data points that are temporally and behaviorally close to the reference samples. This method ensures that positive pairs used in contrastive learning are meaningful in the context of continuous behavioral dynamics.

### A.4.3   IMPROVEMENTS ON SAMPLING BY CONR LOSS

Strictly speaking, our sampling approach is similar to that of CEBRA. We agree with the reviewer that "removing the negative sample batch and replacing it with movement labels does not appear novel." However, our key improvement lies in how we compute the ConR loss, where we determine negative samples supervised by labels. Broadly speaking, we enhance the sampling process by effectively filtering out unintended negative samples from the original set of positive samples extracted by CEBRA. Here's a detailed explanation:

In the CEBRA paper, the authors state: For a continuous context variable $c_t$, we can use a set of time offsets $\Delta$ to specify the distribution. Given the time offsets, the empirical distribution $P(c_{t+\tau} \mid c_t)$ for a particular choice of $\tau \in \Delta$ can be computed from the dataset: we build up a set $D = \{t \in [T], \tau \in \Delta : c_{t+\tau} - c_t\}$, sample a $d$ uniformly from $D$, and obtain the sample that is closest to the reference sample's context variable modified by this distance $(c + d)$ from the dataset.

In practice, this involves the following key code snippets from CEBRA's implementation:

```
https://github.com/AdaptiveMotorControlLab/CEBRA/blob/main/cebra/
distributions/continuous.py#L228
```

```
self.data = continuous # continuous movement labels
self.time_delta = time_delta # time offsets
self.time_difference = torch.zeros_like(self.data,
    device=self.device)
# computing label difference d
self.time_difference[time_delta:] = (self.data[time_delta:]
    - self.data[:-time_delta])
# add the d back to c get query (c + d)
query = self.data[reference_idx]
    + self.time_difference[diff_idx]
```

The problem arises when the continuous labels are imbalanced, as is often the case with movement labels. In such scenarios, labels can quickly transition to infrequent values and then revert back to more frequent ones. When this happens, the computed $(c + d)$, which is intended to index positive (or augmented) samples relative to the anchor after a time offset, may inadvertently point to negative samples. This occurs because the nearest neighbor search retrieves a sample whose label is closest to $(c + d)$, but due to label imbalance, this sample might actually belong to a different class (i.e., a negative sample) (Fig 8a).

As a result, some negative samples are inadvertently mixed into the set of positive samples extracted by CEBRA. Our method addresses this issue by filtering out these unintended negative samples through supervised labels during the computation of the ConR loss (Fig 8b).

In summary, although our initial sampling approach mirrors that of CEBRA, during the computation of the ConR loss we improve negative sampling by selecting samples supervised by labels, and we improve positive sampling by filtering out unintended negative samples.

## A.5 PARAMETERS AND HYPERPARAMETERS

All parameters and hyperparameters for our models are presented in the seven main figures, the thirteen supplementary figures, and summarized Table 1. Additionally, since all training was done in Jupyter Notebook, the hyperparameters are also saved there. Please note that the input data for both NMR and CEBRA are identical. For NMR and CEBRA, no validation data is used; instead, an 80/20 split is applied for training and testing. In contrast, the pi-VAE model uses a 60/20/20 split for training, validation, and testing. pi-VAE was executed on a CPU due to issues with an older version of TensorFlow, which is why we did not compare its execution time with that of NMR and CEBRA. The execution time refers to the training or model fitting time and is associated with the following line of code for both CEBRA and NMR:

```
model.fit(neural, continuous_index)
```

The inference time corresponds to the line of code:

```
model.transform(neural)
```

It converts raw neural dynamics into latent dynamics. This operation is performed on a CPU and takes approximately 0.1 seconds, being similar for both models.

Table 1: Parameters and hyperparameter for NMR and CEBRA models. The XY coordinates represent either hand positions (Figures 1 and 7) or velocities (Figures 2–6). Note that hand reaching angles range from 0 to 360 degrees, but the XY coordinates (maxXY) have different units—such as cm/s or m/s—and may represent different metrics like position or velocity. Since we need to sum the absolute distances of the X-coordinate, Y-coordinate, and angle, we multiply the XY coordinates by a scale factor (XY2Z). This means smaller XY coordinates will have a larger magnification, and vice versa. ITR: iterations, BS: batch size, LR: learning rate, TEMP: temperature $\tau$, maxXY: maximum values of X and Y coordinates, XY2Z: magnification ratio of XY coordinates, PCG: precentral gyrus.

| Figure | ITR (1K) | BS | LR | TEMP | maxXY | XY2Z |
|---|---|---|---|---|---|---|
| 1_S1 positions_NMR | 20 | 512 | 0.001 | 0.045 | 13 | 50 |
| 2_M1_NMR | 10 | 512 | 0.001 | 0.07 | 33 | 10 |
| 2_M1_CEBRA | 5 | 512 | 0.001 | 0.08 | 33 | 10 |
| 2_PMd_NMR | 5 | 512 | 0.001 | 0.08 | 33 | 10 |
| 2_PMd_CEBRA | 10 | 512 | 0.001 | 0.08 | 33 | 10 |
| 4_M1_NMR | 5 | 512 | 0.001 | 0.065 | 33 | 10 |
| 4_M1_CEBRA | 5 | 512 | 0.001 | 0.1 | 33 | 10 |
| 4_PMd_NMR | 5 | 512 | 0.001 | 0.065 | 33 | 10 |
| 4_PMd_CEBRA | 5 | 512 | 0.001 | 0.1 | 33 | 10 |
| 5_M1 sort_NMR | 10 | 512 | 0.001 | 0.06 | 0.2 | 2000 |
| 5_M1 sort_CEBRA | 10 | 512 | 0.005 | 0.1 | 0.2 | 2000 |
| 5_M1 unsort_NMR | 10 | 512 | 0.0005 | 0.06 | 0.2 | 2000 |
| 5_M1 unsort_CEBRA | 10 | 512 | 0.0005 | 0.1 | 0.2 | 2000 |
| 6_M1+PMd_NMR | 10 | 512 | 0.0001 | 0.08 | 31 | 10 |
| 6_M1+PMd_CEBRA | 10 | 512 | 0.0001 | 1 | 31 | 10 |
| 7_PCG positions_NMR | 10 | 512 | 0.0001 | 0.06 | 4 | 100 |
| 7_PCG positions_CEBRA | 10 | 512 | 0.0001 | 0.1 | 4 | 100 |

## A.6 DATASETS

We evaluated a total of 68 sessions (1 + 28 + 37 + 1 + 1) in the main results. Additionally, we analyzed an extra session in the supplementary results to assess the generalizability of our model.

### A.6.1 RAT HIPPOCAMPUS DATASET

This dataset was used in Fig 11 (Grosmark & Buzsáki, 2016).

The data will be automatically downloaded in the CEBRA software package from:

`https://crcns.org/data-sets/hc/hc-11/about-hc-11`

This dataset consists of eight bilateral silicon-probe electrophysiological recordings collected from four male Long-Evans rats. We focused on data from Rat Cicero due to its highly imbalanced nature, characterized by extended periods of pausing at the ends of the track. Data processing was performed using CEBRA.

### A.6.2 MONKEY CENTER-OUT REACHING

**Single unit in S1**

This dataset was used in Fig 1 (O'Doherty et al., 2017):1 monkey, 1 session

The data will be downloaded in the CEBRA software package automatically from:

`https://dandiarchive.org/dandiset/000127`

This dataset includes sorted unit spike times and behavioral data from a monkey performing a reaching task with perturbations. In this experimental task, the monkey used a manipulandum to control a cursor while performing delayed center-out reaches. On some trials, a bump was applied to the manipulandum during the center hold phase before the reach. Neural activity was recorded using an electrode array implanted in somatosensory area 2. Data processing was carried out using CEBRA. Notably, the dataset features a high sampling rate (1 ms time bins), resulting in 600 time points per trial.

**Single unit in M1 and PMd**

Eight direction center-out reaching (Fig 23): 2 monkeys, 28 sessions

This data is released accompanying this paper Gallego-Carracedo et al. (2022).

`https://datadryad.org/stash/dataset/doi:10.5061/dryad.xd2547dkt`

This dataset includes behavioral recordings and extracellular neural recordings from the M1, PMd, and S1 regions of monkeys during an instructed-delayed center-out reaching task. Neural data were collected using one or two Utah arrays.

The data are provided in MATLAB format, and we extracted the following information:

tgtDir: Target direction (in radians) for Monkey Chewie and Mihali. idx-goCueTime: The time at which the "go cue" is issued. vel: XY velocities. M1-spikes: Spiking activity for both Chewie 2015 and Chewie 2016 datasets. PMd-spikes: Spiking activity available only for Chewie 2016.

The time bin size is 30 ms, and we extracted all spikes occurring after each "go cue." For both monkeys, we used 40 time bins. The discrete spike counts were smoothed in MATLAB using a Gaussian kernel with a standard deviation of 1.5 and a kernel size of six standard deviations.

All trials and neurons were included in the analysis.

**LFP in M1 and PMd**

Eight direction center-out reaching (Fig 4): 2 monkeys, 28 sessions

The LFP signals are included in previously released datasets. The main difference in data processing compared to earlier studies is the selection of three specific LFP channels. Unlike spike data, there is no need for smoothing, as the LFP signals are inherently smoothed due to their nature.

### A.6.3 MONKEY NATURAL MOVEMENT

**9 x 9 Grid**

This dataset was used in Fig 5 (O'Doherty et al., 2017):1 monkey, 37 sessions

`https://zenodo.org/records/583331`

The behavioral task involved self-paced reaches to targets arranged in a grid, without gaps or pre-movement delay intervals. We analyzed data from all 37 sessions recorded from monkey 1 ("Indy")

over approximately 10 months. The number of electrodes used in each session varied, with either 96 or 192 electrodes depending on whether one or two arrays were implanted.

For each channel, five vectors were provided: one containing the event times of unsorted spikes and the other four containing sorted spike events. At most, four single units could be recorded simultaneously. The number of unsorted events was significantly larger than the number of sorted events. Occasionally, some channels were empty.

We extracted smoothed firing rates from the spike counts using the same parameters as in the center-out reaching tasks. Since the raw data did not include hand velocities or reaching angles, we computed this information using the provided figure positions and target positions.

**Random target locations**

This dataset was used in Fig 6 (Lawlor et al., 2018): 1 monkey, 1 session

`https://crcns.org/data-sets/motor-cortex/pmd-1/about-pmd-1`

This dataset includes extracellular recordings and behavioral data from a monkey performing a sequential reaching task, designed to examine the roles of the PMd and M1 regions. In the experiment, the monkey controlled an on-screen cursor and was rewarded for moving the cursor to an indicated target, which could be located anywhere on the screen. Each trial consisted of four targets presented sequentially, and there were minimal kinematic constraints for the reaching movements. As a result, the monkey typically executed a relatively smooth series of reaches.

We used data from the first session of Monkey MM, who performed 496 trials of the reaching task. The recordings included 67 neurons from M1 and 94 neurons from PMd. Since this data originates from the same lab that conducted the center-out reaching task, the processing of spike counts and the extraction of movement labels (e.g., velocities) were carried out using similar methods.

### A.6.4 Paralyzed Patient Attempted Movement

Human Handwriting (Fig 7) (Willett et al., 2021): 1 patient, 1 session

`https://datadryad.org/stash/dataset/doi:10.5061/dryad.wh70rxwmv`

We used data collected on 2019.06.03 (1020 days after trial start), which involved attempted handwriting of straight lines similar to the center-out reaching tasks performed by monkeys. Neural recordings were obtained from two 96-channel microelectrode arrays (Utah arrays) implanted in the hand knob area of the precentral gyrus, resulting in raw neural signals with a dimensionality of 192.

In this task, the paralyzed patient was instructed to write short, medium, and long straight lines in 16 directions (as opposed to eight directions used for monkeys). Each direction had 24 repetitions, and we used all of these neural signals. Since there are no real supervised movement labels, we used hand velocities recorded while the patient attempted to write the character "l" which closely resembles a straight line.

### A.7 Mathematical details

#### A.7.1 Problem definition

Consider a training dataset consisting of $N$ examples, which we denote as $\{(x_i, y_i)\}_{i=0}^N$, where $x_i \in \mathbb{R}^d$ is a neural data input, and $y_i \in \mathbb{R}^{d'}$ is its corresponding label. Here, $d$ is the dimension of the input—that is, the raw dimensionality of neural signals or dynamics—which could be the number of simultaneously recorded single units, multi-units, or electrode channels. The value of $d'$ is the dimension of the supervised continuous movement labels. In this study, $d' = 3$, comprising either velocity or position along the X-coordinate, velocity or position along the Y-coordinate, and angle. For the hippocampus dataset, the labels include the animal's location and two values indicating left and right direction. For the movement labels, their distribution $\mathcal{D}_y$ deviates significantly from a uniform distribution (highly imbalanced).

Each neural data sample $x_i$ is passed through the feature encoder $\mathcal{E}(\cdot)$ to obtain the neural embedding $v_i \in \mathbb{R}^{d''}$, where the dimensionality of the latent dynamics $d''$ is predetermined. The objective is to

train a feature encoder $\mathcal{E}(\cdot)$ such that the embeddings $v_i$ are organized both spatially and temporally to correspond to their labels $y_i$.

### A.7.2 LABEL PREDICTION

Let $v_i \in \mathbb{R}^d$ represent the latent embedding for the $i$-th data point, where $d''$ is the dimensionality of the embedding space (e.g., $d'' = 3$). Each data point has a corresponding target label $y_i = [x_i, y_i, \theta_i]$, where $x_i$ and $y_i$ are the spatial coordinates in a 2D space, and $\theta_i$ is the orientation or angle associated with the data point. Thus, $y_i \in \mathbb{R}^3$ represents the $X$-coordinate, $Y$-coordinate, and angle.

A linear regression model is trained to map the neural embeddings $v_i$ to the corresponding labels $y_i$. The model performs the mapping as:

$$\hat{y}_i = W v_i + b, \tag{1}$$

where $W \in \mathbb{R}^{3 \times d}$ is the weight matrix learned by the linear regression model, $b \in \mathbb{R}^3$ is the bias vector, and $\hat{y}_i \in \mathbb{R}^3$ is the predicted label, consisting of the predicted $X$-coordinate, $Y$-coordinate, and angle.

The parameters $W$ and $b$ are learned by minimizing the mean squared error (MSE) between the predicted labels $\hat{y}_i$ and the ground truth labels $y_i$:

$$\min_{W,b} \frac{1}{N} \sum_{i=1}^{N} \|y_i - (W v_i + b)\|^2, \tag{2}$$

where $N$ is the number of data points in the batch. Once trained, the model predicts the labels $\hat{y}_i = [\hat{x}_i, \hat{y}_i, \hat{\theta}_i]$ by applying the learned linear mapping to the embeddings $v_i$.

### A.7.3 CONR LOSS

Let $d(\cdot, \cdot)$ denote the distance measure between two labels. For each anchor sample $i$, the positive samples are those that satisfy $d(y_i, y_p) < \hat{d}$, the negative samples are those that satisfy $d(y_i, y_n) > \hat{d}$ and $d(\hat{y}_i, \hat{y}_n) < \hat{d}$, where $\hat{d}$ is the median of all pairwise distance shown in Fig 1e.

Let's denote $v_i$, $v_p$, and $v_n$ as the neural embeddings of corresponding true labels of $y_i$, $y_p$, and $y_n$. $N_i^+$ is the number of positive samples, $N_i^-$ is the number of negative samples. $K_i^+ = \{v_p\}_p^{N_i^+}$ is the set of embeddings from positive samples, $K_i^- = \{v_n\}_n^{N_i^-}$ is the set of embeddings from negative samples. $sim(\cdot, \cdot)$ is the similarity measure between two feature embeddings (e.g. negative $L_2$ norm). For each anchor $i$ whose neural embedding is $v_i$, true label is $y_i$, and loss is:

$$\mathcal{L}_{\text{ConR}_j} = \frac{1}{N_i^+} \sum_{v_j \in K_i^+} - \log \frac{\exp(sim(v_i, v_j)/\tau)}{\sum_{v_p \in K_i^+} \exp(sim(v_i, v_p)/\tau) + \sum_{v_n \in K_i^-} S_{i,n} \exp(sim(v_i, v_n)/\tau)} \tag{3}$$

where $\tau$ is a temperature hyperparameter and $S_{i,n}$ is a pushing weight for each negative pair shown in Fig 1f:

$$S_{i,n} = \frac{1}{p_{d(y_i, y_n)}} exp(d(y_i, y_n)e) \tag{4}$$

where $\frac{1}{p_{d(y_i, y_n)}}$ is the inverse frequency of labels distances distribution shown in Fig 1c. Since the movement labels have different units, such as centimeters or meters for spatial coordinates and degrees for angles, $e$ is a hyperparameter used to scale the label distance. The absolute value of $S_{i,n}$ depends on the units of the movement labels. While we use the exponential label distance $exp(d(y_i, y_n)e)$, similar results can also be achieved using a linear label distance $d(y_i, y_n)e$. When using the linear distance, the $e$ hyperparameter will typically have a larger value to compensate for the lack of exponential scaling. In this study, we utilized exponential distance, with $e$ set equal to the temperature hyperparameter $\tau$.

The final loss is the summed and averaged loss $\mathcal{L}_{\text{ConR}_j}$ over all anchors $i$ in a batch:

$$\mathcal{L}_{\text{ConR}} = \frac{1}{2N} \sum_{j=0}^{2N} \mathcal{L}_{\text{ConR}_j} \tag{5}$$

### A.7.4 ConR versus InfoNCE loss

For each anchor $i$ whose neural embedding is $v_i$, the InfoNCE loss used by CEBRA is:

$$\mathcal{L}_{\text{InfoNCE}_j} = -\log \frac{\exp(sim(v_i, v_j)/\tau)}{\sum_{n=1}^{N} \exp(sim(v_i, v_n)/\tau)} \tag{6}$$

where $v_i$, $v_j$, and $v_n$ are the anchor, positive, and negative samples, respectively. There are four major differences between the InfoNCE loss in CEBRA and the ConR loss in NMR:

First, only one positive sample $v_j$ is used in ConR loss, instead of multiple $v_j$ that belong to $K_i^+$.

Second, in InfoNCE loss, negative samples are drawn from the entire batch of samples, whereas in ConR loss, only negative samples from $K_i^-$ are selected.

Third, InfoNCE loss does not include a regularizer for negative samples, while ConR loss uses $S_{i,n}$ as a regularizer (to apply pushing weights).

Fourth, InfoNCE loss does not require labels, whereas ConR loss requires labels $y_i$ and $y_n$ for both the anchor and negative samples.

Together, our simplified loss function does not introduce any additional hyperparameters.

### A.7.5 A Theoretical Perspective of Two Losses

For **easy negatives**, where the similarity $sim(v_i, v_n) \approx 0$, their contribution to the denominator of InfoNCE loss (Equation 6) becomes:

$$\exp\left(sim(v_i, v_n)\right) \approx \exp(0) = 1 \tag{7}$$

This means that easy negatives have a minimal impact on the denominator:

$$\sum_{n=1}^{N} \exp\left(sim(v_i, v_n)\right) \tag{8}$$

and are effectively ignored during optimization.

In ConR loss, the contribution of easy negatives is amplified by the weight $S_{j,n}$ (Equation 4).

- If $p_d(y_j, y_n)$ is small (infrequent labels), $S_{j,n}$ becomes large.
- If $d(y_j, y_n)$ is large (far in label space), $S_{j,n}$ is further amplified by the exponential term.

Thus, in ConR loss, the denominator:

$$\sum_{n=1}^{N} S_{j,n} \exp\left(sim(v_i, v_n)\right) \tag{9}$$

ensures that easy negatives contribute more significantly to the optimization.

For **hard negatives**, where $sim(v_i, v_n) \approx sim(v_i, v_j)$, their contribution to the denominator of InfoNCE loss is large:

$$\exp\left(sim(v_i, v_n)\right) \gg 1 \tag{10}$$

This can cause the denominator to dominate, making the numerator:

$$\exp\left(sim(v_i, v_j)\right) \tag{11}$$

relatively small. As a result, the InfoNCE loss becomes small, leading to overlap between hard negatives and positive samples in the latent space.

In ConR loss, $S_{j,n}$ reduces the dominance of frequent hard negatives by scaling their contribution. For hard negatives:

- If $p_d(y_j, y_n)$ (label frequency) is high, $S_{j,n}$ becomes small, reducing their contribution.
- If $d(y_j, y_n)$ is small (close in label space), the exponential term $\exp\left(d(y_j, y_n)e\right)$ does not amplify $S_{j,n}$.

This ensures that hard negatives do not overwhelm the numerator.

Two losses can also be **interpreted probabilistically**. For InfoNCE loss, it minimizes the negative log-probability of correctly identifying the positive $v_j$ given the anchor $v_i$:

$$p_{\text{InfoNCE}}(v_j|v_i) = \frac{\exp\left(\text{sim}(v_i, v_j)\right)}{\sum_{n=1}^{N} \exp\left(\text{sim}(v_i, v_n)\right)} \tag{12}$$

In datasets with an imbalance between positives and negatives:

- Majority negatives dominate the denominator.
- This causes $p_{\text{InfoNCE}}(v_j|v_i) \to 0$, leading to **collapse**, where positives and negatives become indistinguishable in the latent space.

In ConR loss, the probability is modified by the weights $S_{j,n}$:

$$p_{\text{ConR}}(v_j|v_i) = \frac{\exp\left(\text{sim}(v_i, v_j)\right)}{\sum_{n=1}^{N} S_{j,n} \exp\left(\text{sim}(v_i, v_n)\right)} \tag{13}$$

- For **easy negatives**, $S_{j,n}$ amplifies their contribution, ensuring they are not ignored.
- For **hard negatives**, $S_{j,n}$ reduces their dominance, ensuring they do not overwhelm the denominator.

This reweighting in ConR loss prevents collapse and balances the contributions of positives and negatives, leading to better separation in the latent space.

### A.8    CNN AND LSTM

#### A.8.1    CNN ENCODER

NMR used the same feature encoder $\mathcal{E}(\cdot)$ as CEBRA, referred to as the *offset10-model*, where "10" indicates the time offset (number of time bins). This encoder consists of five 1D convolutional layers (`Conv1d`) and is structured as follows:

- `nn.Conv1d(num_neurons, num_units, 2)`: 1st layer, kernel size = 2
- `nn.GELU()`: Activation function
- `cebra_layers._Skip(nn.Conv1d(num_units, num_units, 3), nn.GELU())`: 2nd layer, kernel size = 3, with skip connection
- `cebra_layers._Skip(nn.Conv1d(num_units, num_units, 3), nn.GELU())`: 3rd layer, kernel size = 3, with skip connection
- `cebra_layers._Skip(nn.Conv1d(num_units, num_units, 3), nn.GELU())`: 4th layer, kernel size = 3, with skip connection
- `nn.Conv1d(num_units, num_output, 3)`: 5th layer, kernel size = 3

Here, `num_neurons` refers to the number of input neurons, while `num_units` and `num_output` represent the dimensionality of the hidden layers and the final output latent space, respectively. Gaussian Error Linear Unit (GELU) activation functions are applied after each layer except the final one.

#### A.8.2    LSTM ENCODER AND DECODER

The end-to-end fully supervised dimensionality reduction and motor decoding based on LSTM consists of four major components.

The **first** part involves defining a sequential time-series of neural data to be fed into the LSTM. The objective is to create multiple copies of the neural data `X_train`, each delayed by one time-bin relative to the previous, with the total number of copies equal to the sequence length (we used 10 in this study). Below is the Python function used to create these sequences:

```
def create_sequences(X, y, seq_length):
    X_seq, y_seq = [], []
    for i in range(len(X)):
        X_seq.append(X[i:i+seq_length])
        y_seq.append(y[i+seq_length - 1])
    return np.array(X_seq), np.array(y_seq)

X_train, y_train = create_sequences(neural,
    continuous_index, sequence_length)
train_dataset = TensorDataset(X_train, y_train)
train_loader = DataLoader(train_dataset,
    batch_size=batch_size, shuffle=False)
```

The **second** part is to define the LSTM decoder, which first extracts the latent dynamics using Py-Torch's built-in LSTM layer. Similar to the previously mentioned linear regression decoder, the latent representation is passed through a fully connected layer to decode the movement labels. The LSTMDecoder generates both the predicted labels and the latent dynamics. The third training part uses only the predicted labels, while the fourth part uses only the latents. Below is the implementation of the LSTMDecoder:

```
class LSTMDecoder(nn.Module):
    def __init__(self, input_size, hidden_size,
            num_layers, output_size):
        super(LSTMDecoder, self).__init__()
        self.lstm = nn.LSTM(input_size, hidden_size, num_layers)
        self.fc = nn.Linear(hidden_size, output_size)

    def forward(self, x):
        h0 = torch.zeros(self.num_layers, x.size(0),
            self.hidden_size).to(device)
        c0 = torch.zeros(self.num_layers, x.size(0),
            self.hidden_size).to(device)
        out, _ = self.lstm(x, (h0, c0))  # LSTM output
        latent = out[:, -1, :]  # latents from last time step
        output = self.fc(latent)  # predicted labels
        return output, latent

model = LSTMDecoder(input_size, hidden_size,
        num_layers, output_size).to(device)
```

The **third** part involves training the model to match the predicted labels (outputs) with the ground truth labels (y_batch) by minimizing the MSE loss between the two (loss = criterion(outputs, y_batch)). **In this approach, the modification of the latents occurs indirectly through the training process.** This is fundamentally different from NMR, where the latents are explicitly manipulated, and the training focuses directly on the latents. The training code is as follows:

```
criterion = nn.MSELoss()
optimizer = optim.Adam(model.parameters(), lr=learning_rate)

for epoch in range(num_epochs):
    model.train()
    train_loss = 0.0
    for X_batch, y_batch, _ in train_loader:
        X_batch = X_batch.to(device)# [64bs, 10seq, 65neurons]
        y_batch = y_batch.to(device)# [64bs, 2X&Y]
        optimizer.zero_grad()
        outputs, _ = model(X_batch)  # Latents are not needed
        loss = criterion(outputs, y_batch)
```

```
        loss.backward()
        optimizer.step()
        train_loss += loss.item() * X_batch.size(0)
```

This training approach ensures that the LSTM indirectly adjusts the latent dynamics through optimization on the output predictions, distinguishing it from NMR's direct manipulation of the latents.

The **fourth** part involves extracting the latent dynamics using the previously trained best-performing model. This step does not require predicted labels but focuses solely on the extracted latents. The process is as follows:

```
model.load_state_dict(torch.load('M1_best_model.pth'))
model.eval()

latent_list = []
with torch.no_grad():
    for X_batch, _, _ in train_loader:
        X_batch = X_batch.to(device)
        _, latents = model(X_batch) # No prediction here
        latent_list.append(latents.cpu().numpy())

latents = np.vstack(latent_list)
```

Here, the trained model is loaded using `load_state_dict`, and `torch.no_grad()` is used to disable gradient computation, optimizing memory and computation during inference. The extracted latents from each batch are appended to `latent_list` and then vertically stacked (`np.vstack`) to form the final latent representation.

A.9   DECODING CROSS MULTI-SESSIONS

For training model cross multiple sessions, the model needs to be trained separately for each session or animal. We achieved this by iterating through all datasets in the designated folder. In the uploaded Jupyter Notebooks, all .ipynb files that have "batch" in their name indicate they are designed for multi-session training. For example, the file "Fig2_NMR_SU_Batch_PMd.ipynb" trains the NMR model on all neural data from PMd.

In the Fig 3 and Table 2, which presents the cross-session decoding experiment. In this experiment, the only fine-tuning performed was rotating the angles of the latent dynamics. This was necessary because, in some sessions, the angles of the extracted latent dynamics were rotated by 45 degrees or flipped relative to the ground truth movement trajectories. To address this, we used the orthogonal Procrustes method from SciPy:

```
https://docs.scipy.org/doc/scipy/reference/generated/scipy.linalg.
orthogonal_procrustes.html
```

Using this method, we selected a target angle and rotated the entire 3D latent dynamics with the computed orthogonal matrix. This alignment preserves local details and the relative positions of each reaching direction. Once all latent dynamics were aligned with their corresponding movements, we trained a linear regression decoder on 80% of the training data from one session and used it to decode movements in other sessions with the 20% held-out test data.

A.10   BENCHMARK OF MOTOR DECODING

Table 3 presents the motor decoding performance (explained variance) on the $9 \times 9$ Grid Random Target Task (RTT). The methods NDT1, NDT2, EIT, and POYO are all based on transformer architectures. The NDT2 paper mentioned that "A 10% test split is used in each evaluation session." This study used the session from 20160407, which is the first session in our 37 sessions. The results reported in Table 3 came from their Fig 3A, where single-session transformers trained from scratch for NDT1/2 are represented by brown/blue dots, respectively.

The POYO paper mentioned that "for every session, we hold out 20% of the trials for testing." This study used the session from 20170202, which is just one day after the last session in our 37 sessions.

Table 2: Datasets information for decoding across sessions, hemispheres, animals, and years related to Fig 3. n/a: no recordings in the PMd of right hemisphere.

| Date | Monkey | Hemisphere | Trial | M1 | PMd |
|---|---|---|---|---|---|
| 140217 | Mihili | Right | 208 | 44 | 104 |
| 140218 | Mihili | Right | 225 | 38 | 121 |
| 140303 | Mihili | Right | 208 | 52 | 66 |
| 140304 | Mihili | Right | 203 | 39 | 76 |
| 140306 | Mihili | Right | 217 | 43 | 86 |
| 140307 | Mihili | Right | 216 | 26 | 66 |
| 150313 | Chewie | Right | 1038 | 86 | n/a |
| 150309 | Chewie | Right | 1026 | 72 | n/a |
| 150629 | Chewie | Right | 179 | 49 | n/a |
| 150630 | Chewie | Right | 178 | 44 | n/a |
| 160929 | Chewie | Left | 208 | 74 | 114 |
| 161005 | Chewie | Left | 202 | 82 | 167 |
| 161006 | Chewie | Left | 209 | 63 | 192 |
| 161007 | Chewie | Left | 168 | 70 | 137 |
| 161014 | Chewie | Left | 740 | 88 | 190 |
| 161021 | Chewie | Left | 286 | 84 | 211 |

Therefore, we reported our last day's 80% training and 20% testing data in Table 3. Note that this session in POYO paper was not excluded but was not available from the website.

Note that we did not directly compare pretrained models with NMR and other models trained from scratch, as it would be unfair; the main difference is likely due to the pretrained dataset rather than the network structure.

| Method | 9 x 9 Grid Random Target |
|---|---|
| Wiener Filter | 0.5438 |
| GRU | 0.5951 |
| MLP | 0.6953 |
| AutoLFADS + Linear | 0.5931 |
| NDT1 + Linear | 0.5895 |
| NDT1-Sup | 0.4621 |
| NDT1 (Ye et al., 2023) | 0.5174 |
| EIT | 0.4691 |
| NDT2 (Ye et al., 2023) | 0.5189 |
| POY0 | 0.6850 |
| NMR 80% train | 0.8151, 0.8175 ± 0.0451 |
| NMR 20% test | 0.7144, 0.7107 ± 0.0422 |

Table 3: Behavioral decoding results of hand velocities. Most results are taken from the Table 3 of POYO paper (Azabou et al., 2023). NDT2 results are extracted from Fig 3A of the NDT2 paper (Ye et al., 2023), as the raw values were not provided. Our NMR results come from Fig 5e (sorted units). The ± symbol indicates the standard error. The Wiener Filter employs multiple linear filters, GRU and AutoLFADS are based on recurrent neural networks, MLP is based on a feedforward neural network, and all the remaining methods are based on transformer architectures. Two rows in NMR: the first value represents the last session among the 37 sessions that matches the previous results, while the second and third values represent the mean and standard deviation across all 37 test sessions.

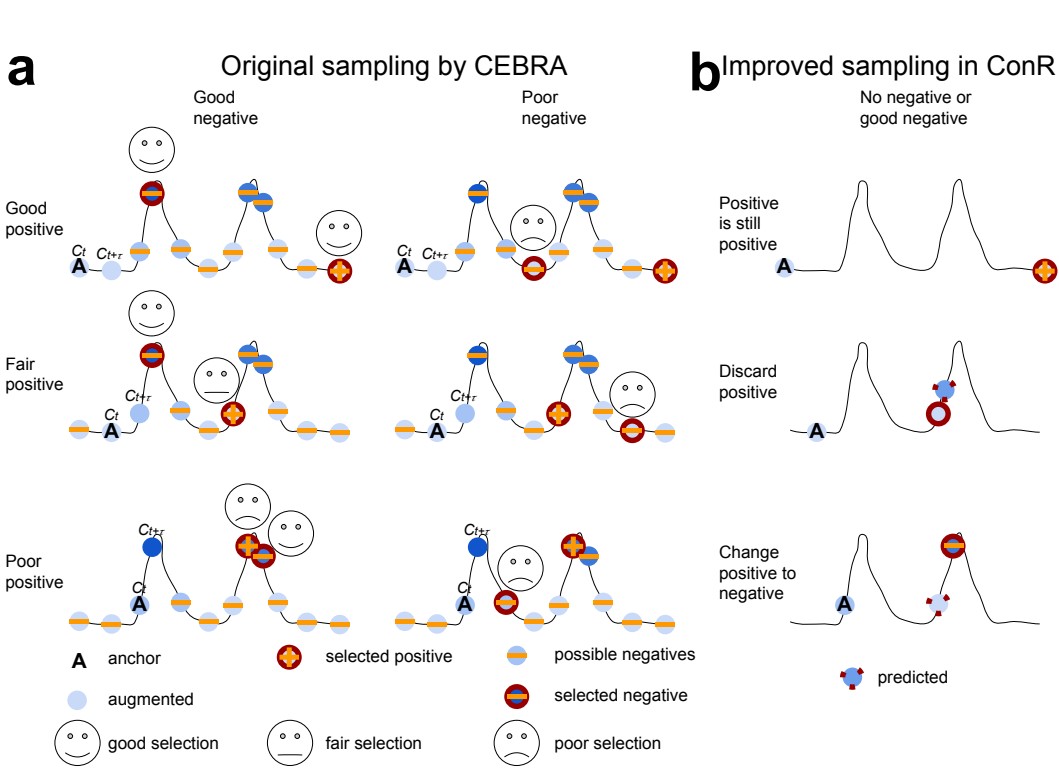

Figure 8: Original and Improved Sampling Strategies. **a** Depending on the position where the anchor ("A") sample falls on the continuous label $C_t$, the label of the augmented sample $C_{t+\tau}$ could be similar (first row), slightly different (second row), or very different (third row). The positive sample ("+") will have a label that is closest to the augmented sample. Therefore, if the augmented sample changes rapidly relative to the anchor sample, the selection of positive samples may be fair or poor. Since negative sampling is uniform and unsupervised, negative samples (denoted by "-") could appear in many positions. They might fall into a continuous label that is supposed to be very different from the anchor (first column) or, incorrectly, into a position similar to the anchor (second column). **b** NMR does not require negative samples for computing the ConR loss. Instead, it refines the original positive samples extracted by CEBRA. There are three situations: 1) No change needed: If the originally selected positive sample is close to the anchor within a distance threshold (first row). 2) Discarded: If it is far away from the anchor and the predicted label is also far away. 3) Changed to negative sample: If the true label is far from the anchor but predicted to be close. For simplicity in visualization, we show only one anchor, one negative, and one positive sample. In the actual experiment, there are 512 samples for each type. In the computation of the ConR loss, each anchor sample is compared with 512 positive samples to classify those samples into positive, negative, or discarded.

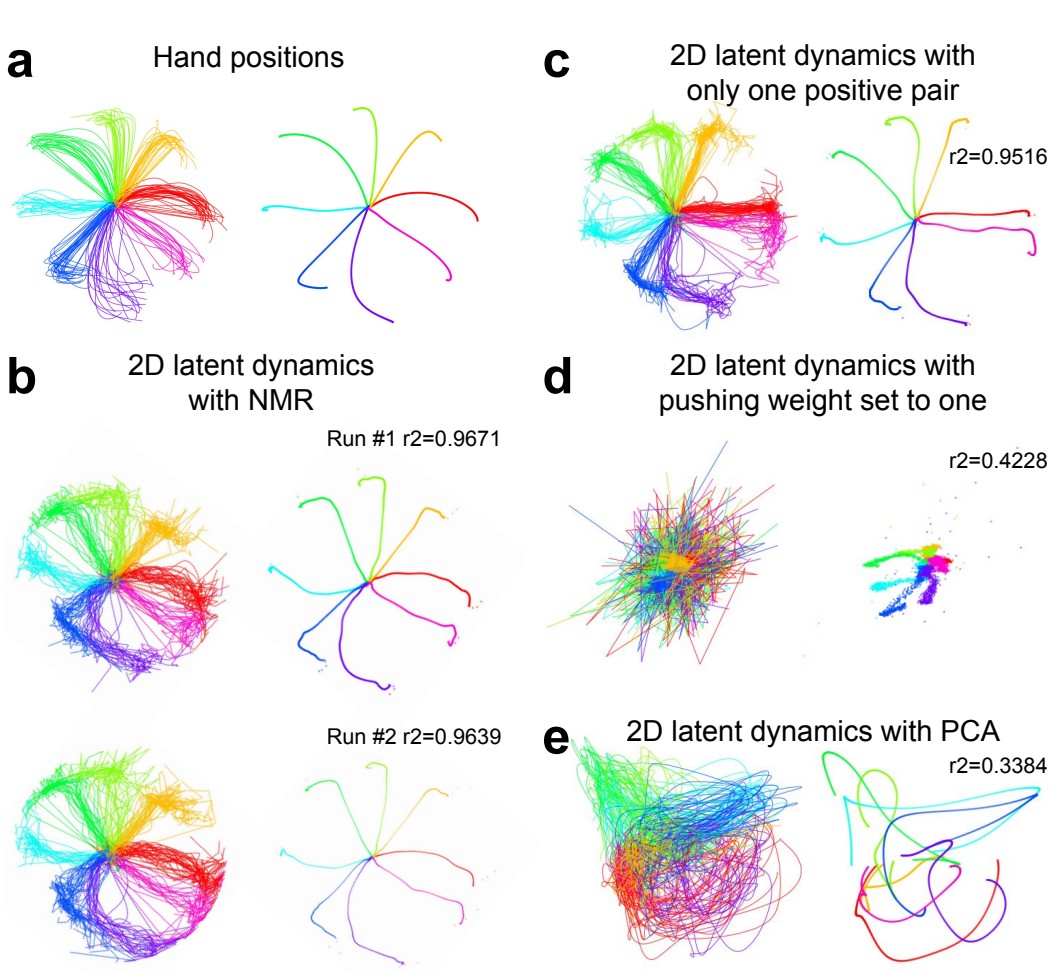

Figure 9: Single-trial and trial-averaged hand positions and latent dynamics. **a** Ground truth movement trajectories. **b** Two additional examples of Fig 1b (right panel). **c** 2D latent dynamics extracted with NMR, containing only one positive pair (i.e., its own augmented sample with the same label and zero distance to the anchor sample). **d** 2D latent dynamics extracted with NMR, with the pushing weight $S_{i,n}$ set to one. **e** 2D latent dynamics were extracted using PCA applied to the raw 65-dimensional neural signals. The explained variance represents the decoding performance of a linear regression decoder applied directly to the raw 65-dimensional neural signals.

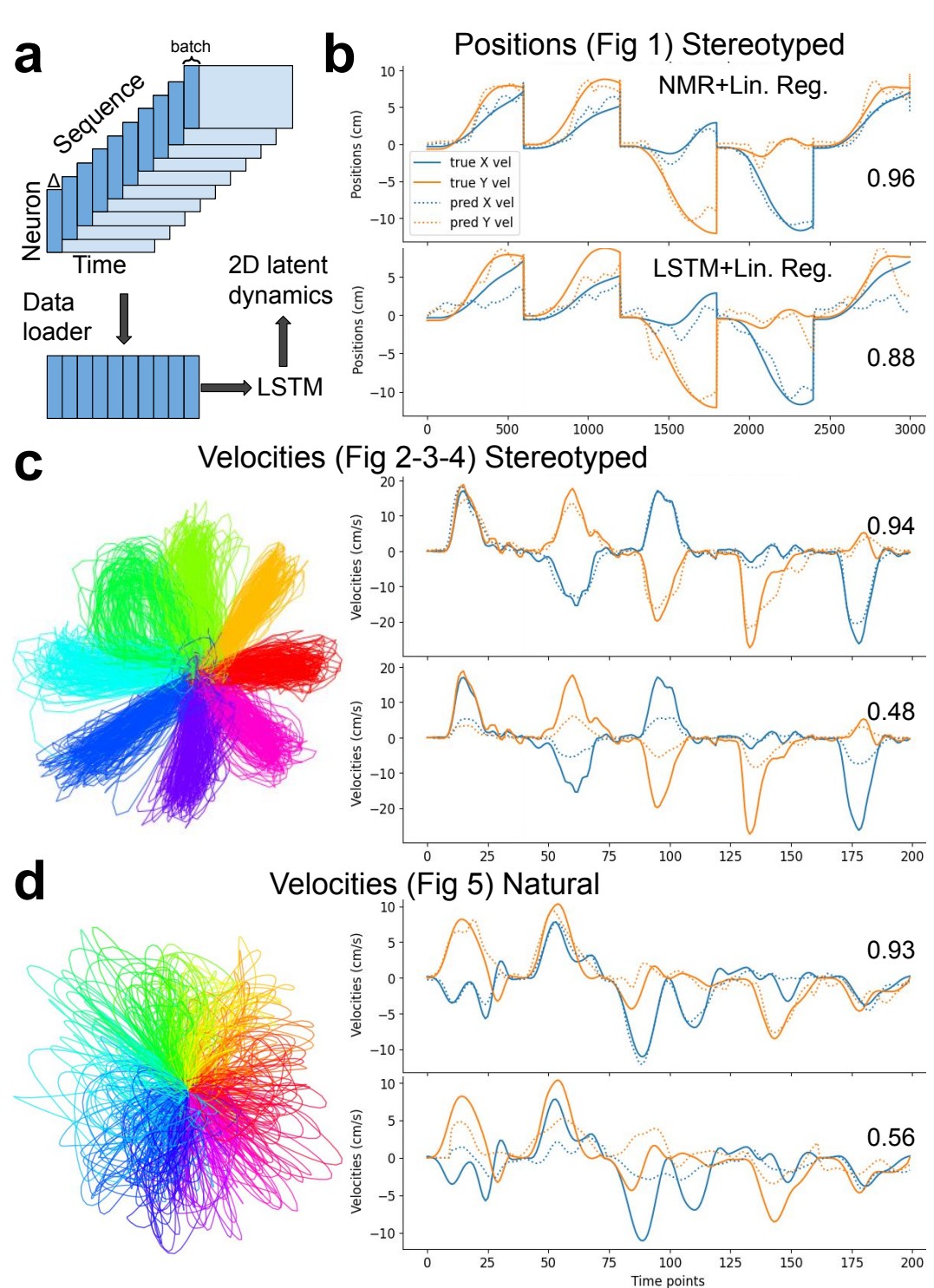

Figure 10: Supervised decoding based on latent dynamics extracted using a long short-term memory (LSTM) model compared to NMR. **a** Data sampling strategy utilized for training the LSTM and extracting the latent dynamics. **b-d** Comparison of ground truth (solid lines) and predicted (dashed lines) movement trajectories for X (blue) and Y (orange) coordinates. Predictions are generated using a linear regression decoder applied to 2D latent dynamics extracted by NMR (top) and LSTM (bottom). Numerical annotations indicate the explained variance of the movements.

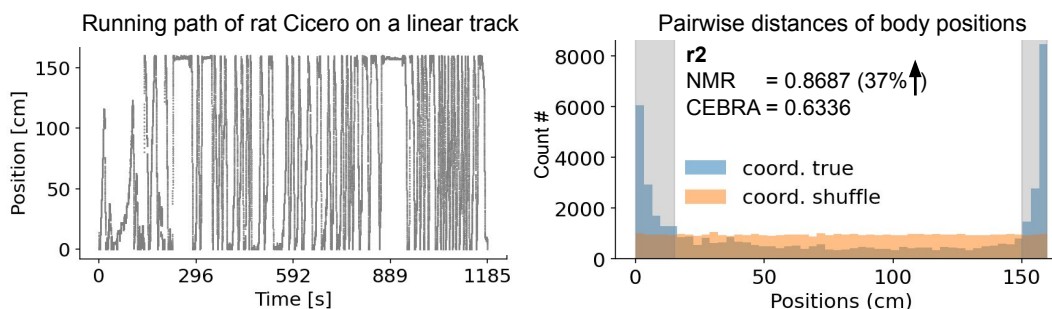

Figure 11:    Model evaluation was conducted on body movement tasks using neural data from free-moving rats. Left: Multi-channel electrophysiological recordings were collected while a rat traversed a 1.6 m linear track either "leftward" or "rightward." It is worth noting that the rat occasionally paused at the ends of the track, resulting in data imbalance. Right: Pairwise distances between samples from the entire body movement trajectories reveal a highly imbalanced distribution. At both the beginning (0–15 cm) and the end (150–160 cm) of the track, significantly more data were collected compared to the shuffled condition (beginning: 27% vs. 9%; end: 27% vs. 6%). Two-dimensional latent dynamics were extracted using NMR and CEBRA and employed to predict movements through linear regression. The explained variance ($r^2$) of body movements was 37% higher with NMR compared to CEBRA. Both models were optimized via hyperparameter searches for learning rate and temperature.

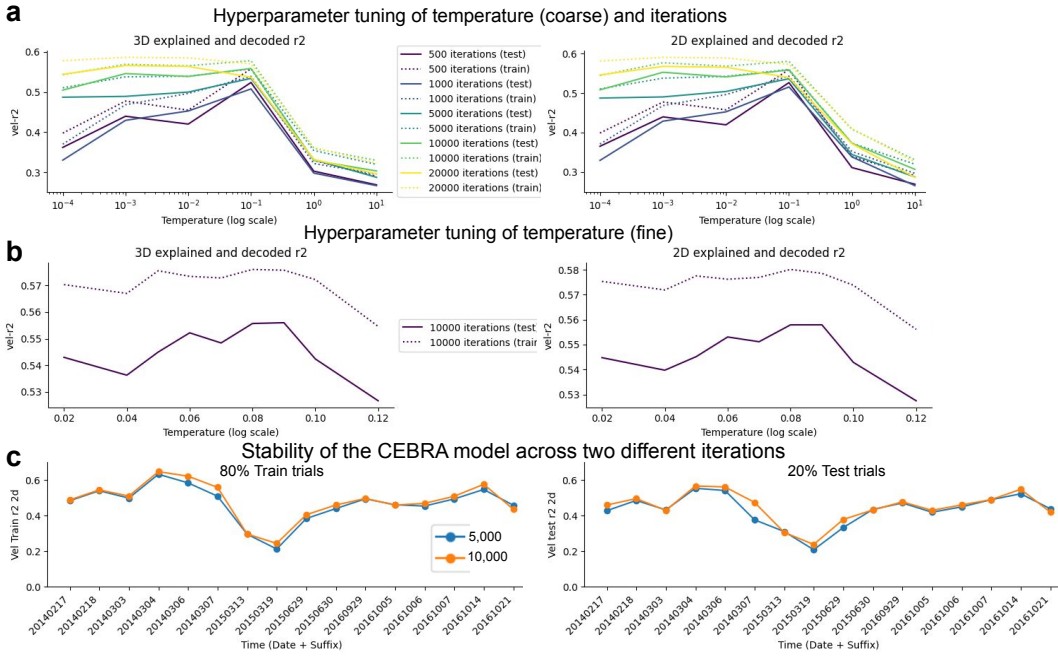

Figure 12: Hyperparameter tuning and stability of CEBRA. **a**. Hyperparameter search across five different iterations and six different temperatures. The evaluated session is from Monkey C (20161014, M1). **b**. A finer hyperparameter search at 10,000 iterations. **c**. Explained variance (left) and decoded variance (right) at two different iteration numbers across 14 sessions in M1.

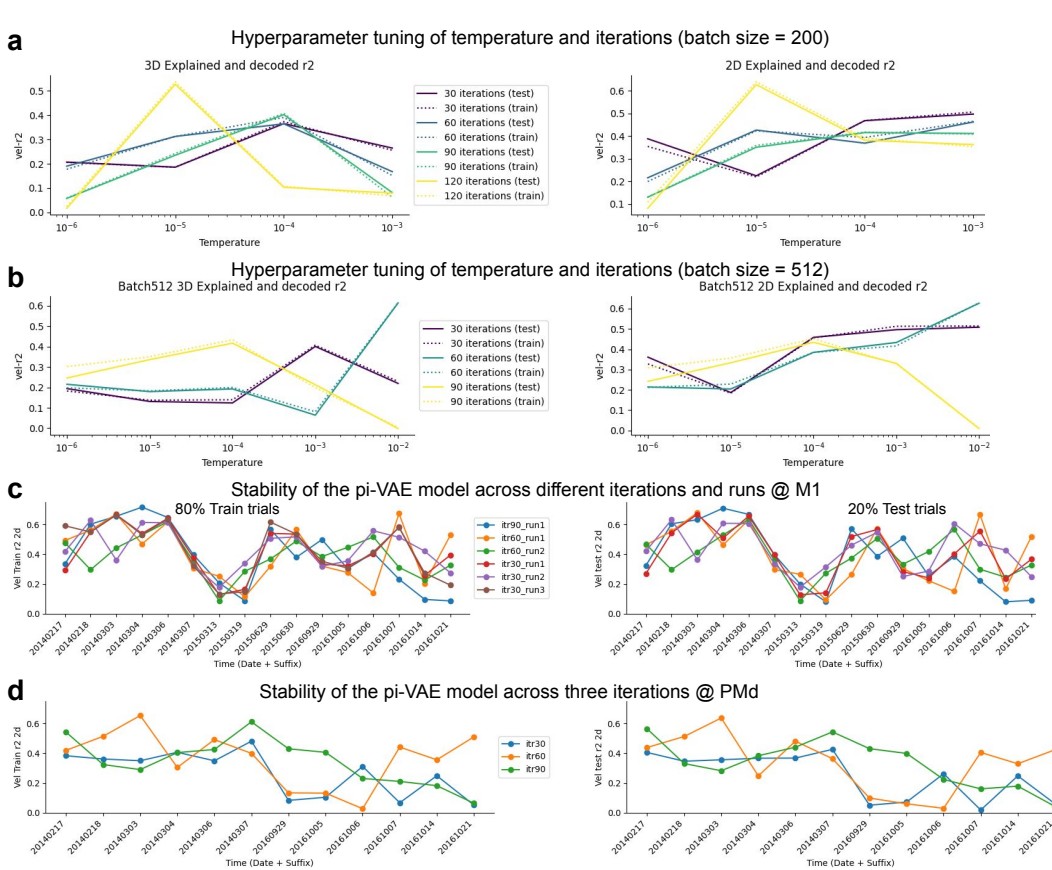

Figure 13: Hyperparameter tuning and stability of pi-VAE. **a**. Hyperparameter search across four different iterations and four different learning rates. The evaluated session is from Monkey C (20161014, M1). **b**. Similar search, but using a larger batch size. **c**. Explained and decoded variance under different iteration numbers and across multiple runs. Note that the performance shows a similar trend across sessions but has larger variability within each session. **d**. Similar to panel c, but models are evaluated in PMd.

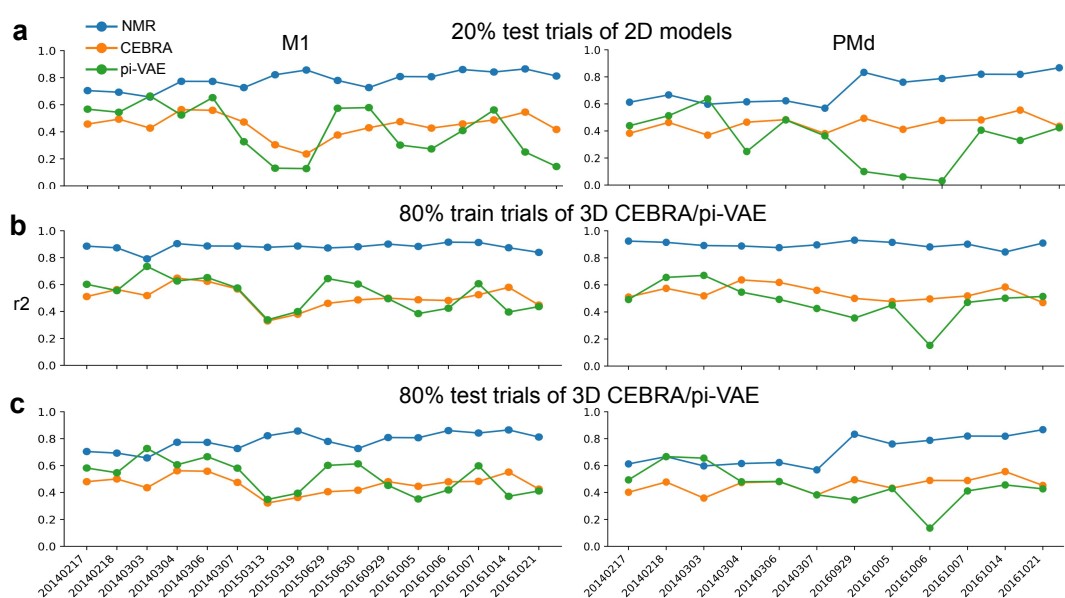

Figure 14: Test trial performance and 3D model comparison. Same format as Figure 2, but for held-out 20% test trials using 3D CEBRA and pi-VAE models. **a**. Decoded r² across sessions in M1 and PMd using 2D models. **b**. Explained r² and **c**. Decoded r² for 2D NMR compared to 3D CEBRA and 3D pi-VAE models.

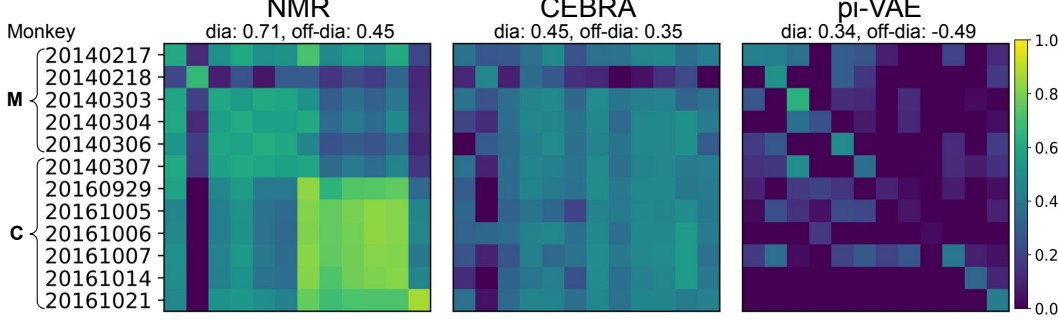

Figure 15: Decoding results in PMd, following the same format as Fig 3. The t-statistics and p-values for the diagonal values are 10.1821 and 1.9e-06 (NMR vs CEBRA), 5.0372 and 1.1e-03 (NMR vs pi-VAE), 1.8407 and 0.2783 (CEBRA vs pi-VAE). The t-statistics and p-values for the off-diagonal values are 6.5845 and 3.0e-09 (NMR vs CEBRA), 6.2945 and 1.3e-08 (NMR vs pi-VAE), 5.7219 and 2.0e-07 (CEBRA vs pi-VAE).

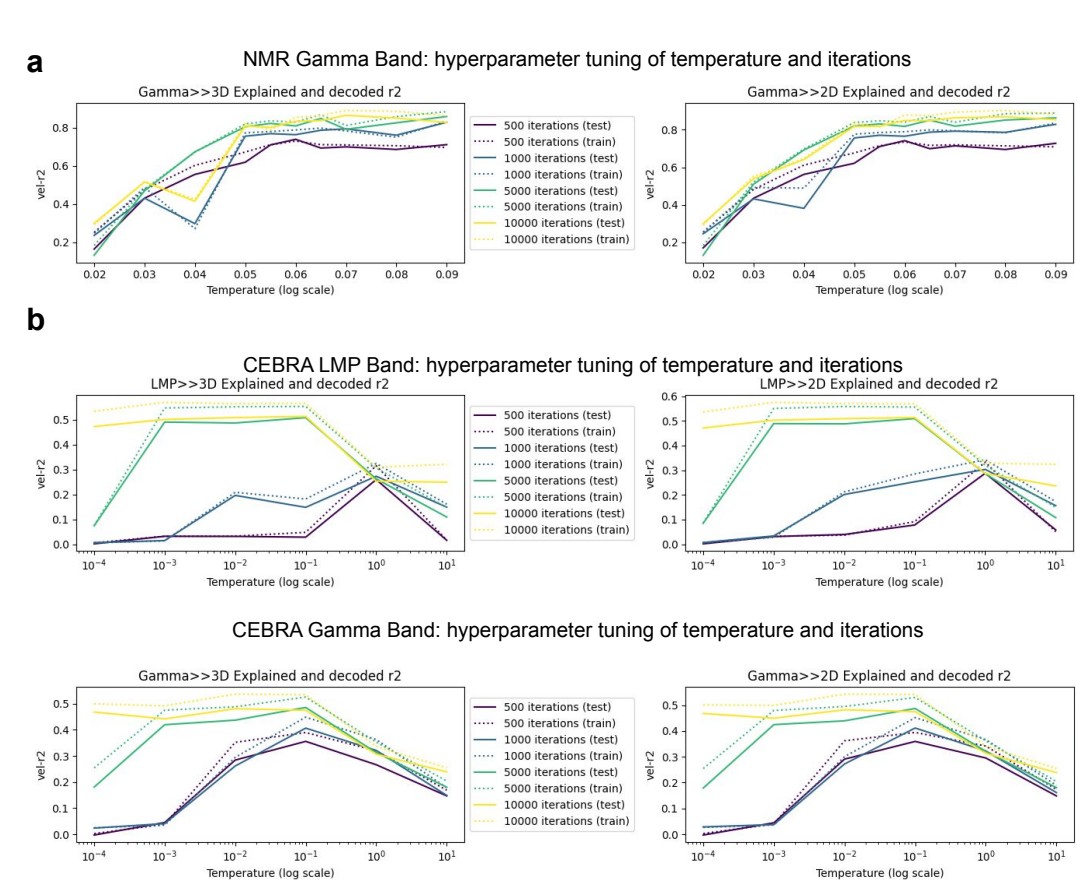

Figure 16: Hyperparameter tuning for two models. **a**. Explained variance across four iterations and eight temperatures in the high Gamma band (200-400 Hz) for NMR. **b**. Similar tuning results for CEBRA in the LMP (smoothed LFP signals) and Gamma bands.

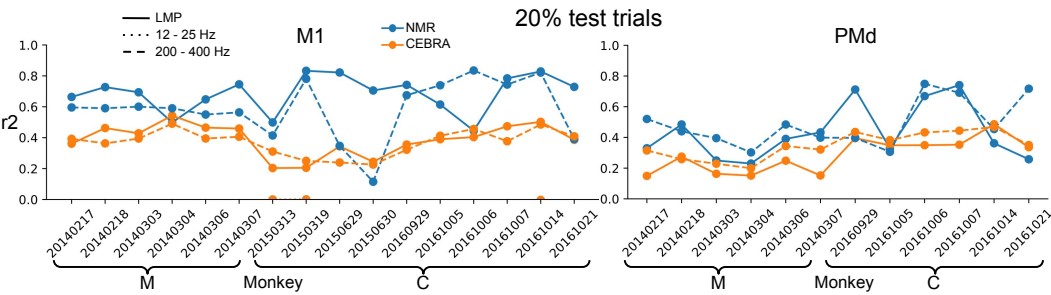

Figure 17: Decoding performance on held-out test trials, following the same format as Fig 4.

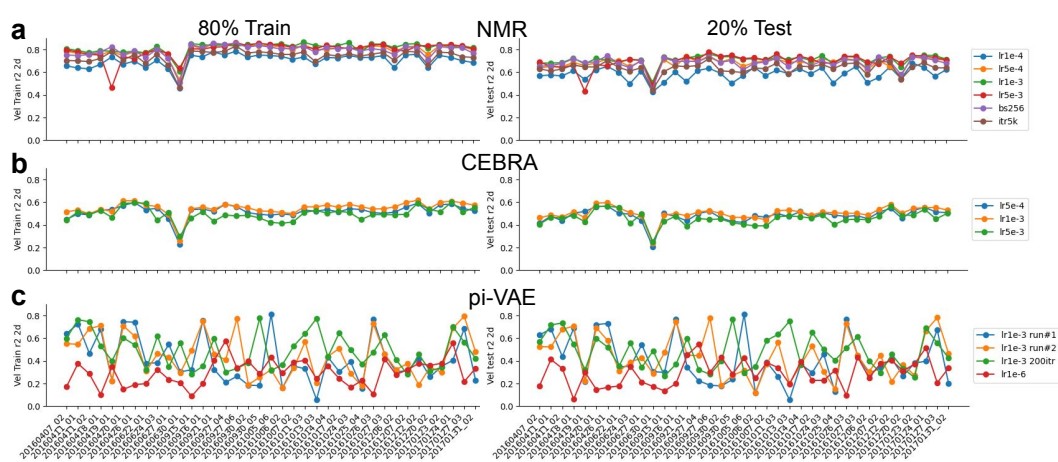

Figure 18: Explained variance under different hyperparameters. **a**. Variance results for four different learning rates, smaller batch sizes (256 vs. 512), and fewer iterations (5,000 vs. 10,000). **b**. Results for three different learning rates. **c**. Comparison between two runs using the same learning rate but higher iterations (200 vs. 100) and a much lower learning rate.

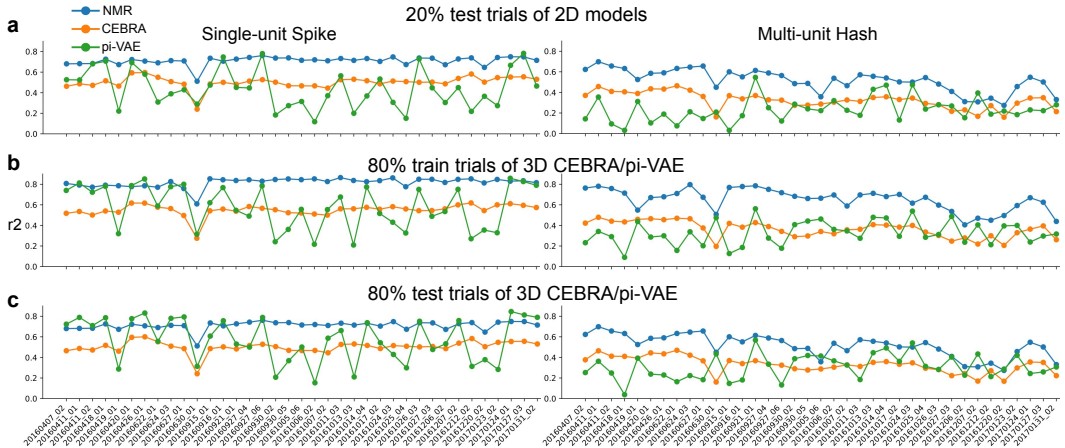

Figure 19: Decoding performance for test trials (**a**) and 3D CEBRA/pi-VAE models (**b**, **c**).

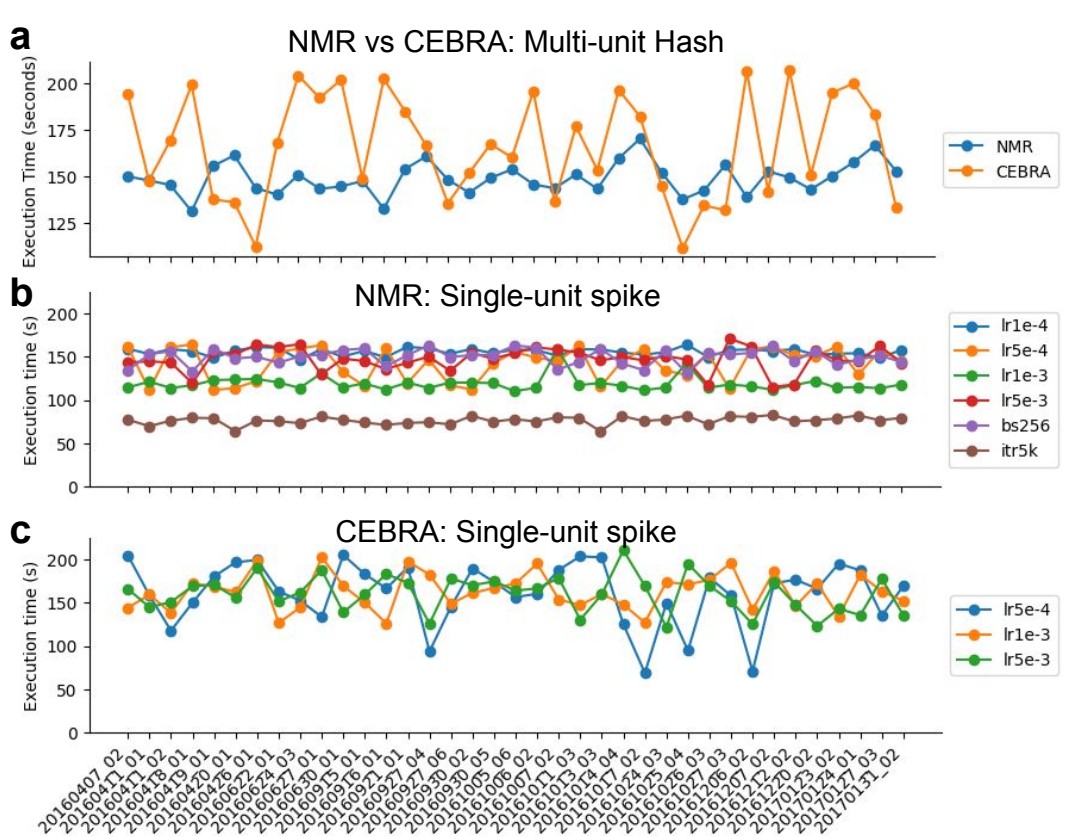

Figure 20: Execution time for NMR and CEBRA models. **a**. Same format as Figure 5f, but for unsorted events. **b**. Comparison of execution times for four different learning rates, smaller batch sizes, and fewer iterations. **c**. Execution time results for three different learning rates.

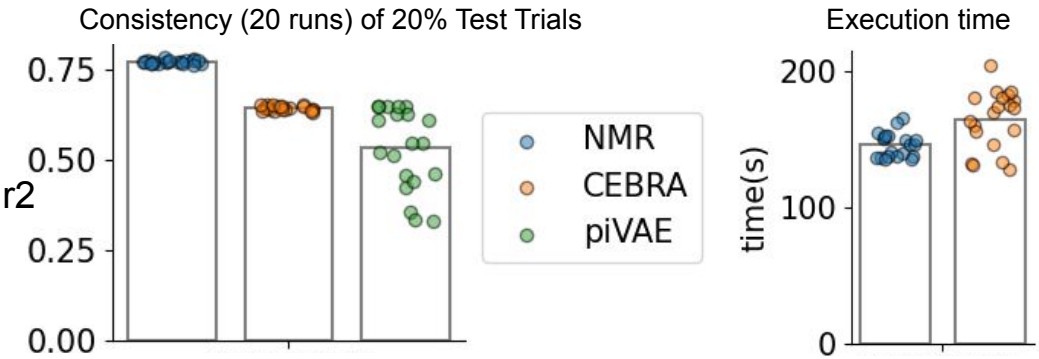

Figure 21: Model decoding performance in the testing trials and execution times.

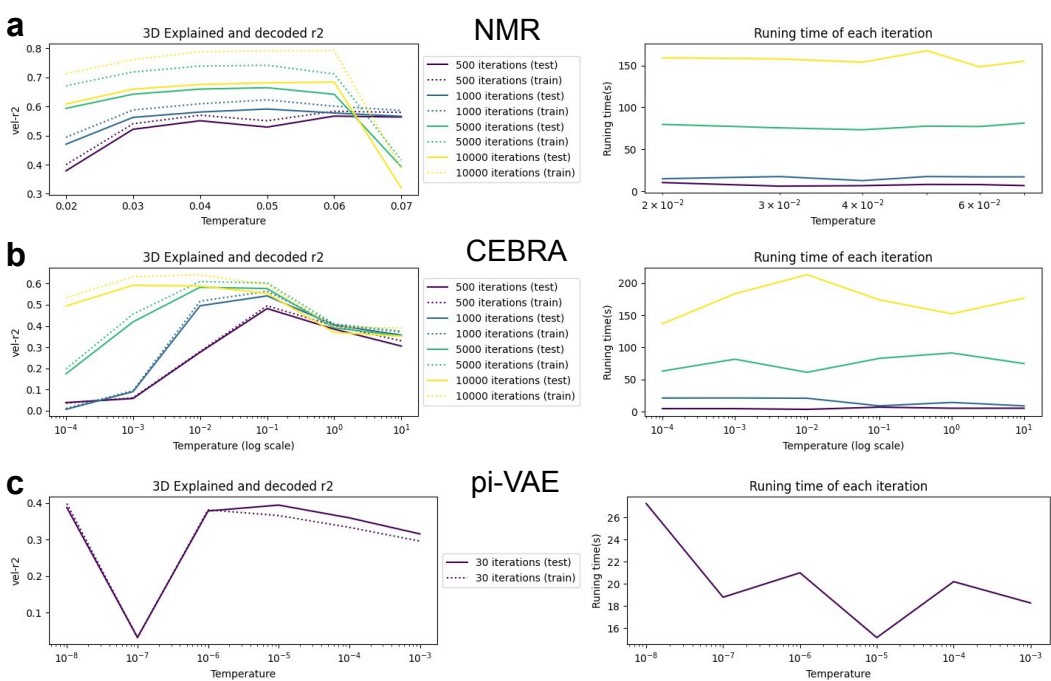

Figure 22: Hyperparameter search and runtime of NMR (**a**), CEBRA (**b**), and pi-VAE (**c**) models

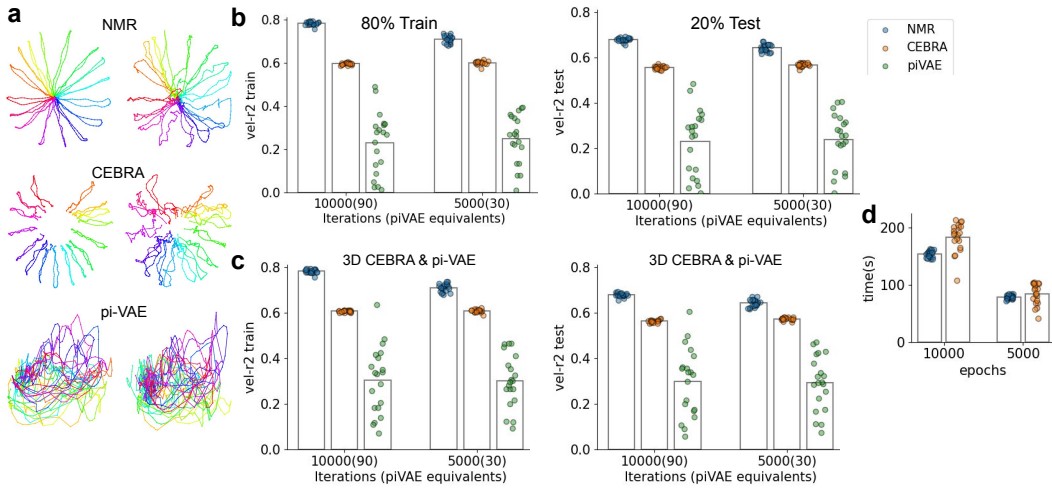

Figure 23: 2D latent dynamics of the three models and performance across different conditions. **a**. 2D latent dynamics in training trials (left) and held-out test trials (right). **b**. Explained variance of hand velocities in training and test trials at two sets of iterations. **c**. Similar analysis for 3D CEBRA and pi-VAE models. **d**. Execution time comparison between NMR and CEBRA at two different iteration levels.

