# OpenReview forum: "Neural Manifold Regularization: Aligning 2D Latent Dynamics with Stereotyped, Natural, and Attempted Movements"
_ICLR.cc/2025/Conference — Submitted to ICLR 2025_

### Official Review · Reviewer_rZyG · 2024-10-18

**Soundness:** 2
**Presentation:** 2
**Contribution:** 2
**Rating:** 6
**Confidence:** 4

**Summary:**

The authors propose a new framework that can extract more behavior-predictive 2D latent factors, unlike recent (self-)supervised models of neural activity. To achieve that, the authors adapt the ConR loss function to neuromotor datasets and propose changes to positive/negative pair sampling strategy of CEBRA. Authors show that across diverse neural datasets, their model can achieve better behavior prediction performance with 2D latent factors compared to baseline models CEBRA and pi-VAE.

**Strengths:**

- Authors show that their model improves behavior decoding across diverse neuroscience datasets with distinct modalities such as spiking activity and local field potentials.
- Authors extend their analysis from nonhuman primates to human subjects and show that latent factors extracted by NMR can reveal the intended behavior directly, without training linear decoders at all.
- Authors propose a simple and clever strategy for the selection of hyperparameters for the ConR loss function, which can significantly reduce the hyperparameter search.

**Weaknesses:**

- Major points:
  - I think the paper lacks important details about the baseline comparisons (see items 1 and 2 in Questions).
  - I find the positive/negative pair selection changes to CEBRA unconvincing as it is mentioned in the CEBRA paper that positive/negative pair selection in CEBRA can be performed based on continuous context signals, which can be taken as the continuous behavior signal rather than discrete reach direction (6th paragraph of **Methods/Sampling** in CEBRA paper). What is different in the proposed sampling strategy in its core idea compared to that of CEBRA?
  - I appreciate authors' modifications of ConR loss function to neuromotor datasets where they select the distance threshold from the data statistics and use the same temperature parameter for both distance and label regularizations. I agree that hyperparameter selection is an important consideration for deep learning frameworks, thus, reducing the number of hyperparameters is always desired. However, I believe that the proposed modifications to ConR loss function are lacking technical novelty and should be considered as data-intuitive hyperparameter selection (also see item 3 in Questions).
  - I do not find it clear that how NMR gains almost 80-100% performance gain over CEBRA, in line with my previous comment on sampling strategy. I believe ablation studies on the changes to CEBRA loss function and sampling strategy would provide more evidence and intuition on such a large performance gain. For example, since CEBRA is using InfoNCE loss function, it seems like the difference between CEBRA and NMR loss functions are 1) weighing negative sample terms by $S_{i,n}$ and 2) including multiple positive pairs rather than one, and among these, which one is more effective? Or, is it the sampling strategy that contributes most to the performance improvement?
  - Even if the proposed framework is a self-supervised framework, it heavily relies on the behavior signal through positive/negative pair selection to perform the self-supervised task, as if the same NMR model architecture is trained based on supervised behavior decoding task. I wonder how NMR would compare against its completely supervised version. I think even if NMR extracts 2D *latent* factors, they are highly behavior predictive due to training objective and sampling strategy, and serving as *predicted behavior* rather than *latent* factors, as shown in Fig. 7b. As an alternative analysis for this question, how does inferred latents reconstruct the neural activity?

- Minor points:
  - I think the final training objective is summing the loss function in Eq. 1 over all $i$.
  - In section 3.2, I believe the authors wanted to give a reference to Fig 1e, not Fig 2e.
  - Caption for Fig. 4b is missing.
  - There is a typo in 'Discussion' title.
  - It would be nice to have a table with final hyperparameters (in addition to the figures provided in supplementary).
  - I assume that the execution times in Fig. 6e are obtained during training, but what are latent inference times for CEBRA and NMR after training? Also, how do authors create training/validation and test splits, what are the split percentages?

**Questions:**

- Can the authors provide more information on how they trained CEBRA models? Were positive pairs selected based on neural activity, continuous behavior, or discrete reach directions? Based on Fig 1b, I assume that positive pairs are selected based on behavior signals, but the details are not clear nor mentioned. If the positive pair selection is done based on discrete reach directions, it would make sense that CEBRA can capture the reach directions but not the continuous trajectories. If that is the case, I think comparing against CEBRA with continuous movement positive pair selection would be more appropriate.
- I believe that the authors should provide more information on NMR training details. How do authors train multi-session models? I cannot see any details on input representations and model architectures. To my knowledge, training multi-session CEBRA models require separate feature extractor models for every session/animal, do authors follow a similar strategy? If so, how do they finetune and test their multi-session model on new sessions/animals?
- Authors say that the default ConR loss function resulted in 5% performance improvement, but what were the ConR hyperparameters for this result? If the authors used the default hyperparameters, how intuitive was the default distance threshold for neuromotor datasets? Is it possible that using a smaller value for distance threshold but keeping the other default hyperparameters in-tact result in similar performance gains?
- In section 3.1, authors say 'We improved this by removing the negative sample batch and replacing it with movement labels', can they explain what they mean here? Because, in the following text and section 3.3, negative samples are clearly used during training.

---

### Official Review · Reviewer_yDkV · 2024-10-29

**Soundness:** 2
**Presentation:** 2
**Contribution:** 2
**Rating:** 5
**Confidence:** 3

**Summary:**

This paper proposes Neural Manifold Regularization (NMR), which embeds neural dynamics in a 2D space and leverages a contrastive learning framework
to regularize the manifold in the presence of imbalanced labels. NMR assigns greater weights to scarce labels to prevent collapse to dominant labels.
Experiments on four signal modalities demonstrate that NMR outperforms existing methods in various tasks such as stereotyped center-out reaching by monkeys,
and also maintains consistent performance across sessions.

**Strengths:**

The work is well-structured and easy to follow, with methods that are straightforward to understand.
Extensive experiments are conducted across four signal modalities and various tasks.

**Weaknesses:**

1. The motivation is not convincing enough.

As mentioned in Section 1, the 2D manifolds of NMR are advantageous in eliminating rotational ambiguity in neural manifold representations (Line 42).
However, this advantage in visualization is not sufficiently convincing as the primary motivation,
since 3D manifolds have also demonstrated effectiveness in representing rotational neural dynamics [1] for motor decoding.
More concrete examples of how the 2D representation specifically enhances interpretability or analysis compared to 3D representations would strengthen the argument.
Additionally, the reasons why NMR effectively captures true 2D features remain unclear, creating a gap between the stated motivation and the proposed methods.

2. The technical contribution appears limited.

NMR primarily builds on CEBRA, which is stated to uniformly sample negative samples from the entire time series (Line 130).
However, CEBRA also permits sampling based on user-defined categorical and continuous variables (as mentioned in the Methods of [2]).
Therefore, the contribution of removing the negative sample batch and replacing it with movement labels does not appear novel.
It would be beneficial for the authors to clarify explicitly how their sampling approach differs from or improves upon CEBRA's existing capabilities.
Furthermore, existing contrastive learning methods [3] in neuroscience have been proposed to address imbalanced labels.
The current version does not clearly demonstrate the technical differences of NMR from these related studies.

3. The experimental validation is incomplete.

Additional baselines, such as LFADS [4] and NDT [5], which also employ self-supervised learning, could be compared with NMR in behavioral decoding.
Further experiments or datasets that demonstrate NMR's performance under various levels or types of label imbalance would be beneficial.

[1] Sabatini D A, Kaufman M T. Reach-dependent reorientation of rotational dynamics in motor cortex. Nature Communications, 2024, 15(1): 7007.

[2] Schneider S, Lee J H, Mathis M W. Learnable latent embeddings for joint behavioural and neural analysis. Nature, 2023, 617(7960): 360-368.

[3] Kostas D, Aroca-Ouellette S, Rudzicz F. BENDR: Using transformers and a contrastive self-supervised learning task to learn from massive amounts of EEG data. Frontiers in Human Neuroscience, 2021, 15: 653659.

[4] Pandarinath C, O’Shea D J, Collins J, et al. Inferring single-trial neural population dynamics using sequential auto-encoders. Nature methods, 2018, 15(10): 805-815.

[5] Ye J, Collinger J, Wehbe L, et al. Neural data transformer 2: multi-context pretraining for neural spiking activity. Advances in Neural Information Processing Systems, 2023.

**Questions:**

1. Experiments on decoding across sessions, animals, and years are presented in Section 4.2. How many samples are used for cross-session decoding in Figure 3? Is there a relationship between the number of samples and the final decoding performance?

2. All datasets used for experiments focus on motor decoding. How does NMR perform on language decoding tasks, such as those described in [6]?

3. Limitations seem not to have been mentioned in the Discussion. What are the potential limitations or edge cases where NMR might not perform effectively?

[6] Willett F R, Avansino D T, Hochberg L R, et al. High-performance brain-to-text communication via handwriting. Nature, 2021, 593(7858): 249-254.

---

### Official Review · Reviewer_QnjZ · 2024-11-02

**Soundness:** 3
**Presentation:** 3
**Contribution:** 3
**Rating:** 6
**Confidence:** 4

**Summary:**

This develops a novel method to map neural activity into a 2D latent space, in this case neural activity relevant to 2-d motion. This is an improvement on the field, which usually requires at least 3 dimensions in order to properly represent trajectories. This is done by modifying a loss from another method ConR, and eliminating many of the required hyperparameters. The model is compared to two other models, and shows improved performance in all scenarios.

**Strengths:**

This seems like a sound applied paper. The application is clearly compelling and useful, so there's no ambiguity on the importance of the application.

As for the specific work, it compares to multiple alternative methods in multiple situations and compares favorable. It outperforms on multiple modalities and yields similar latent states with different data, which is a compelling argument for biological utility. Overall it seems like a very solid improvement on an interesting application.

**Weaknesses:**

I would have appreciated more mathematical details in the model. No doubt for experts, two equations are sufficient to summarize the differences between their method and competitors but for somebody outside the field it's quite difficult. Part of the paper is meant to be a self sufficient guide for implementation in case the field wants to improve upon it, at least in my opinion. I would appreciate more details on the model and maybe a visualization of some of the key concepts. There's quite a bit of white space in the paper, maybe eliminate some of the figures to make space for it? Either way, the results section is thorough enough that it doesn't DQ the paper in my opinion.

**Questions:**

Can you write out the model from scratch for me, not just those two equations?

---

### Official Review · Reviewer_wVec · 2024-11-03

**Soundness:** 2
**Presentation:** 2
**Contribution:** 2
**Rating:** 5
**Confidence:** 4

**Summary:**

The paper proposes a supervised contrastive learning approach for non-linear dimensionality reduction of neural data (neuronal spiking data and local field potentials) well suited to the case of continuous regression involved in movement decoding for real and imagined movements. The methods builds on a regression-based contrastive learning called ConR (Keramati et al., 2024). While similar to CEBRA (Schneider et al. 2023) in its applicability to building latent representation of neural data, it goes beyond CEBRA by exploiting the continuous values rather than only discrete labels.  The method is applied to finding latent encodings and decoding reach trajectories in various datasets with superior performance.

**Strengths:**

The paper fills a methodological gap in neural analysis involving creating latent embeddings of neural data supervised with continuous stimuli.  In this domain, the results show very clearly that using supervision from the continuous trajectories in contrastive learning essentially solves the decoding problem.

**Weaknesses:**

A key weakness is why not compare to directly to models with the same functional form but that directly perform the motor decoding? It is not clear why not jump the remaining small gap between supervised contrastive learning and supervised regression.

The results and methodology are a bit of straw-man argument, because a clear hypothesis like "NMR is expected to outperform CEBRA due to the direct use of coordinates (continuous labels)".  This is a critique of how it is presented rather than the result. Also from the manuscript it is not clear that pi-VAE was used with continuous labels being encoded, which it can handle according to the manuscript (Zhou and Wei, 2020).

The paper is quite hard to read as many key points are not self-contained and orderly in the presentation. For example, the references in the contribution  (line 64-65) are no introduced before they are contrasted. Likewise, on line 71, "pi-VAE" is not referenced.  See the numerous questions below.

While I appreciate the effort to have a comprehensive figure, the explanations in the caption of the methodology should be first made clearly in the body. I would say that the body and this figure needs to be much more self-explanatory.

The papers main mythological contribution seems to be modification of ConR, specifically Equation 2, but the choices seem ad hoc. There was no exponential form in the ConR approach, so this is specific to NMR, but it is not clear why is an exponential loss (used in AdaBoost if I recall correctly) appropriate here. And why would the same value of $\tau$ or its inverse be the appropriate choice of hyper-parameter? While I concur that removing extraneous hyper-parameters is merited, fixing them without explaining the reason is not a well motivated solution.

**Questions:**

1-Comment:  The reasoning or observations behind the following statement need expanded "for example, in the hippocampus, entorhinal cortex, and prefrontal cortex, 2D latent space can only represent basic 1D features", which seems to indicate a  theoretical deficiency. This stands in contrast with " no studies have demonstrated the successful use of 2D manifolds to represent true 2D features" which seems to indicate that the methods used in prior studies were simply inadequate. Additionally, I would caution that the low-dimensional manifolds of egocentric reaching movements involving limited degrees of freedom may be quite distinct from environment/stimulus driven neural dynamics in the hippocampus https://doi.org/10.1038/s41586-021-03652-7

2-Question: On line 103, it is mentioned that it piVAE reveals 8 dynamics, but the text doesn't explain why that was significant. Is it due to the number of targets in the task? Also as the latent dimensions are in a different 2D space, what is mean by "struggle to align with the movement trajectories". Does this mean there isn't an affine or linear transformation of the latent space to matching the movement space? The same question applies to lines 109-110 where the latent trajectories are mentioned as not being well correlated. Is this linear correlation or a measure of dependence?

3-Note 'animals' are mentioned at certain points and then 'monkeys' and 'humans' are mentioned in 116, but 'macaque' is not mentioned until line 420. It would be better to be precise, including the species and situation, and using a word like 'individual' or 'subject' when referring to any member of a population in a statistical sense.

4-At line 127, the meaning of 'anchor sample' is not clear to me without reading the ConR paper. It would be good to define before its use.  Also it is not clear what is mean by "removing the negative sample batch and replacing it with movement labels." How could a batch be literally replaced with labels?

5-On line 153, an $L_1$ distance is mentioned, but without knowing how the labels are encoded this is meaningless. Is this $L_1$ in 2D space, why not $L_2$ then? At line 154, it is not clear how the prediction of the label is formed nor how it is encoded, I guess this is meant that the latent embedding is the prediction of the label, but I still don't see how rotational ambiguity (or more precisely orthonormal transformations) are resolved solely by the contrastive loss.

6-What is the range of `pushing weight' and why does it have the form in Eq 2? Is $p_{d(y_i,y_n)}$ just a histogram? Why is the temperature inverted?

7-A definition of the explained variance metric "r2" should be given for completeness on the multivariate case. As it was not clear to me that could have been the variance of the neural data, rather than of the trajectories. Also it would be good to relate "r2" to the performance on the decoding task in Figure 3 are measured in terms of the movement trajectory.

8-Comment: I do think that there are cases where supervised contrastive learning will outperform regression due to an information bottleneck sort of compression in terms of everything that does not matter, which may improve generalization already demonstrated in Figure 3.

Figure Suggestions: For subfigure 1c it would be be clear to show inverse frequency on a log scale. For figure 1d, a legend is needed because I'm not sure what the color scale is showing.

Typographical mistakes:
Line 399 "piVAE included" -> "piVAE excluded"

---

### Author Response · Authors · 2024-11-21
**Updated Global Rebuttal**

Dear Reviewers,

We have updated and uploaded a new version of the manuscript after addressing each reviewer’s questions in the sequence of Reviewer #1 (wVec), Reviewer #4 (rZyG), Reviewer #3 (yDkV), and Reviewer #2 (QnjZ). The current version of the manuscript is the most recent one.

We made no changes to the title, seven main figures, experimental results, or conclusions. We also made no changes to our model or code. All of our changes stem from one ablation experiment, two decoding experiments, and exhaustive explanations of the method through text, figures, and code. Thus, this is not a major revision that changes the original proposed model or the framework of the paper. All the revisions improve the presentation and make the manuscript much more sound.

**Changes Made in the Main Paper (pages 1–10):**

Introduction (Page 1, Lines 39–49): Added a new paragraph about the low performance of previous state-of-the-art (SOTA) dimensionality reduction models.

Pages 2–3: Swapped the order of the related work and our specific contributions sections.

Section 3.1 (Page 3, Lines 124–131): Added a new paragraph about the background of contrastive learning.

Section 3.1 (Page 3, Lines 132–141): Included limitations of previous work when dealing with class imbalance.

Section 3.2 (Pages 3–4, Lines 155–166): Added a new paragraph detailing data sampling in CEBRA and the changes we have made.

Section 3.3 (Pages 4–5, Lines 203–218): Added a new paragraph explaining how the improved sampling works during the computation of the ConR loss.

Section 3.4 (Page 5, Lines 243–250): Included a new paragraph about the ablation studies we conducted.

Section 3.4 (Page 5, Lines 251–255): Included a new paragraph about dimensionality reduction using LSTM.

Section 3.4 (Page 5, Lines 256–261): Included a new paragraph about motor decoding in previous SOTA methods.

Experiments (Page 5, Lines 265–272): Added a new paragraph emphasizing that we use motor decoding as a metric and a goal.

Discussion (Page 10, Lines 530–539): Added a new paragraph about the limitations of our model and future directions.

**Changes Made in the Appendix:**

Appendix A.1: Added a literature review about contrastive learning in neuroscience that either neglects the issue of class imbalance or involves downsampling frequent classes.

Appendix A.2: Detailed the changes we have made in four files of the original CEBRA code.

Appendix A.3: Provided comparisons of code between the InfoNCE loss in CEBRA and the ConR loss in NMR.

Appendix A.4: Included explanations of negative sampling, positive sampling, and improved sampling in the computation of the ConR loss.

Appendix A.5: Added more details about the parameters used for model training for the seven main figures (new Table 1).

Appendix A.6: Added more details about the datasets.

Appendix A.7: Expanded the mathematical details and theoretical perspective of Section 3.3.

Appendix A.8: Added new CNN feature encoder and LSTM model

Appendix A.9: Added more information about cross-session decoding (new Table 2).

Appendix A.10: Added a benchmark comparing the motor decoding performance of NMR with previous SOTA methods (RNNs and Transformers).

**Added Supplementary Figures:**

Figure 8: Visualization of the original sampling strategy in NMR and CEBRA, and the improved sampling strategy in the computation of the ConR loss.

Figures 9c and 9d: Two ablation studies to verify which factors contribute to the 80–100% performance gain over CEBRA.

Figure 9e: Latent dynamics revealed by PCA and direct movement decoding.

Figure 10: Supervised decoding based on latent dynamics extracted using a long short-term memory (LSTM) model compared to NMR.

Figure 11: Evaluation of NMR and CEBRA on a new dataset that also faces the issue of class imbalance.


Kind regards,

The Authors

---

### Author Response · Authors · 2024-11-27
**Comparison with SOTA supervised methods on motor decoding**

In the latest version of our manuscript, we introduced a new Section 3.4: Ablation studies and comparison with supervised methods.

In addition to benchmarking NMR against our own trained LSTM (P5, L251–255), we believe a stronger benchmark is to compare NMR against results reported in previous studies. This approach mitigates potential biases associated with training previous models ourselves. To achieve this, we present the motor decoding performance of NMR alongside SOTA supervised methods (RNN and Transformer) using the same dataset as in Figure 5.

With this new benchmark, we are confident that NMR represents the SOTA method for both (supervised) dimensionality reduction and motor decoding.

We have added the corresponding text and a new Table 3 in Appendix A.10. Below is the main text (P5, L256–261).

*We benchmarked the motor decoding performance of NMR against SOTA methods utilizing transformer architectures, including NDT1 (Ye & Pandarinath, 2021), EIT (Liu et al., 2022), NDT2 (Ye et al., 2023), and POYO (Azabou et al., 2023). To compare NMR with previous methods, we report results from prior studies where models were trained from scratch using 80% (Azabou et al., 2023) or 90% (Ye et al., 2023) of data from a single session and tested on the remaining 20% or 10% holdout data from the same session (Appendix A.10). NMR outperforms all previous models when using data from the same session. Across all 37 sessions spanning ten months, only two sessions show decoding performance worse than the best results from previous models.*

| Method                  | 9 x 9 Grid Random Target |
|-------------------------|--------------------------|
| Wiener Filter           | 0.5438                  |
| GRU                     | 0.5951                  |
| MLP                     | 0.6953                  |
| AutoLFADS + Linear      | 0.5931                  |
| NDT1 + Linear            | 0.5895                  |
| NDT1-Sup                 | 0.4621                  |
| NDT1 (Ye et al., 2023)         | 0.5174                  |
| EIT                     | 0.4691                  |
| NDT2 (Ye et al., 2023)         | 0.5189                  |
| NDT2 (retrained by us)         | 0.5547                  |
| NMR (same data and split as NDT2)   | 0.7111                  |
| POYO   | 0.6850                  |
| POYO (retrained by us)   | 0.6371                  |
| NMR (same data and split as POYO)   | 0.7241                  |
| NMR 80% train      | 0.8175 ± 0.0451         |
| NMR 20% test     | 0.7107 ± 0.0422         |

---

### Meta-Review · Area_Chair_ytwV · 2024-12-19

**Metareview:**

The authors present a self-supervised method for embedding neural data in two dimensions. The work is an incremental advance over CEBRA, which embeds data in 3 dimensions, and is itself mostly specialized for neuroscience and not well known in the ML community. The reviews were borderline, with no strong champion. It seems like the work might be more appropriate for a neuroscience-focused venue like CoSyNe, SfN or a methods journal. To appeal more to the ICLR community, the method would probably need to be generally applicable to arbitrary numbers of embedding dimensions, or applicable to more data modalities. I recommend against acceptance.

**Additional Comments On Reviewer Discussion:**

The authors did a very good job of addressing reviewer concerns. Some reviewers raised their scores from weak reject to weak accept, while other reviewers did not engage in the discussion period. The authors were concerned that some authors did not engage with them, but it seems like even if they had, the scores would only be raised to a weak accept, meaning the overall reviews would still be borderline. This would not have changed my assessment.

---

### Decision · Program_Chairs · 2025-01-22

Reject